# PCGF5 is required for neural differentiation of embryonic stem cells

Mingze Yao[1,2], Xueke Zhou[1,2], Jiajian Zhou [3], Shixin Gong[1,2], Gongcheng Hu[1,2], Jiao Li [1,2], Kaimeng Huang [1,2], Ping Lai[1,2], Guang Shi[1,2], Andrew P. Hutchins [4], Hao Sun [3], Huating Wang[5] & Hongjie Yao [1,2]

Polycomb repressive complex 1 (PRC1) is an important regulator of gene expression and development. PRC1 contains the E3 ligases RING1A/B, which monoubiquitinate lysine 119 at histone H2A (H2AK119ub1), and has been sub-classified into six major complexes based on the presence of a PCGF subunit. Here, we report that PCGF5, one of six PCGF paralogs, is an important requirement in the differentiation of mouse embryonic stem cells (mESCs) towards a neural cell fate. Although PCGF5 is not required for mESC self-renewal, its loss blocks mESC neural differentiation by activating the SMAD2/TGF-β signaling pathway. PCGF5 loss-of-function impairs the reduction of H2AK119ub1 and H3K27me3 around neural specific genes and keeps them repressed. Our results suggest that PCGF5 might function as both a repressor for SMAD2/TGF-β signaling pathway and a facilitator for neural differentiation. Together, our findings reveal a critical context-specific function for PCGF5 in directing PRC1 to control cell fate.

[1] CAS Key Laboratory of Regenerative Biology, Joint School of Life Sciences, Guangzhou Institutes of Biomedicine and Health, Chinese Academy of Sciences, Guangzhou Medical University, Guangzhou 510530, China. [2] Guangdong Provincial Key Laboratory of Stem Cell and Regenerative Medicine, CAS Center for Excellence in Molecular Cell Science, Guangzhou Institutes of Biomedicine and Health, Chinese Academy of Sciences, Guangzhou 510530, China. [3] Department of Chemical Pathology, Li Ka Shing Institute of Health Sciences, the Chinese University of Hong Kong, Hong Kong 999077, China. [4] Department of Biology, Southern University of Science and Technology, Shenzhen 518055, China. [5] Department of Orthopaedics and Traumatology, Li Ka Shing Institute of Health Sciences, the Chinese University of Hong Kong, Hong Kong 999077, China. Correspondence and requests for materials should be addressed to H.Y. (email: yao_hongjie@gibh.ac.cn)

Polymb repressive complexes (PRCs) have been classified into two major complexes, named PRC1 and PRC2, based on their composition as well as their enzymatic activity toward specific histone residues. PRC2 complex catalyzes histone H3 lysine 27 tri-methylation (H3K27me3) through its core components EZH1/EZH2, EED and SUZ12. PRC1, conversely, contains the core ubiquitin ligase RING1A/B protein, which catalyzes H2AK119ub1, and promotes chromatin compaction and gene suppression[1]. Recent evidence has suggested that H2AK119ub1 may not strictly lead to transcriptional repression, at least during certain stages of development[2,3].

PRC1 can be divided into six major groups defined by their six different PCGF subunits (PCGF1, PCGF2, PCGF3, PCGF4, PCGF5 and PCGF6)[4]. It has been suggested that PRC1 is recruited in a hierarchical manner to sites with pre-existing PRC2 activity and H3K27me3. However, H3K27me3-binding CBX proteins are limited to canonical PRC1 complexes containing either PCGF2 or PCGF4[4], while all PCGF proteins interact with RYBP/YAF2 to form noncanonical PRC1 complexes without CBX proteins[4–6]. De novo recruitment of the noncanonical PRC1 complexes (PCGF1, PCGF3 and PCGF5) results in the formation of a polycomb domain containing PRC2 and H3K27me3[7]. In addition, PCGF5-PRC1-AUTS2 activates gene expression in the mouse central nervous system, suggesting PCGF5 may also play a role in gene activation in a context-dependent manner except the repressive function by PRC1[8].

In this study, we find that PCGF5 is highly expressed and is required for the differentiation of mESCs towards a neural cell fate. Although PCGF5 is not required for mESC self-renewal, its loss blocks neural differentiation by activating SMAD2 phosphorylation and the TGF-β signaling pathway. Small molecule-mediated inhibition of TGF-β signaling pathway or over-expression of PCGF5 can rescue the capability of mESCs to differentiate towards a neural cell fate. PCGF5 executes these effects by promoting histone H2AK119ub1 both in vitro and in vivo in a RING1B-dependent manner. PCGF5 loss-of-function results in reductions of H2AK119ub1 and H3K27me3 at the promoters of TGF-β target genes (such as *Nodal, Lefty1, Lefty2*), upregulation of pSMAD2, during neural differentiation, leading to dysfunction of TGF-β signaling pathway, and a consequent blockage towards a neural cell fate. On the other hand, loss of PCGF5 results in higher levels of H2AK119ub1 and H3K27me3 around neural specific genes and keeps these genes repressed. PCGF5 is not only recruited to repressed genes, but also binds to active genes during neural differentiation, suggesting PCGF5 may function as both a repressor for SMAD2/TGF-β signaling pathway and a facilitator for neural differentiation. Taken together, this study reveals a critical context-specific function and mechanism for PCGF5-PRC1 in controlling neural differentiation of mESCs.

## Results

### PCGF5 is highly expressed in neural stem cells. 
By screening epigenetic factors that are important for neural differentiation of human ESCs (hESCs) from an expression array analysis, we identified *TET2, PCGF4* and *PCGF5* as upregulated in human neural stem cells compared with hESCs (Fig. 1a). TET2 has already been reported to play an important role in differentiation to neuroectoderm[9] and BMI1 (PCGF4) is required for the self-renewal of neural stem cells in the nervous systems in mouse[10]. Therefore, we focused on studying the role of PCGF5 in hESCs and mESCs during neural differentiation, reasoning that PCGF5 might be important in mediating ESC neural differentiation. We induced differentiation of both hESCs and mESCs toward a neural cell fate and confirmed the upregulation of *Pcgf5* (Supplementary Fig. 1a–e). Due to the time-consuming nature of neural differentiation in hESCs (Supplementary Fig. 1a, b), we decided to use the faster mESCs as a model system to investigate PCGF5 function in neural differentiation.

### PCGF5 is dispensable for mESC self-renewal. 
To understand PCGF5 function in mESC pluripotency and differentiation, we knocked out the *Pcgf5* gene in mESCs using the transcription activator-like nucleases (TALEN) targeting the second exon of *Pcgf5* (Fig. 1b). The cells were double selected by puromycin/G418 and single colonies were picked and cultured for further analysis. *Pcgf5* knockout in mESCs was verified by qRT-PCR, Western blot, genomic PCR and Southern blot (Fig. 1c and Supplementary Fig. 1f, g).

Loss of PCGF5 in mESCs did not impact the protein level of pluripotency markers OCT4 and NANOG (Fig. 1d), did not alter mESC proliferation (Supplementary Fig. 1h, i) nor affect the expression of a panel of pluripotency genes (Supplementary Fig. 1j, k). Therefore, we conclude that PCGF5 is not required for mESC self-renewal, which is consistent with the lower expression of *Pcgf5* in undifferentiated ESCs than neural stem cells.

### PCGF5 loss-of-function blocks mESC neural differentiation. 
We next induced mESCs to differentiate towards a neural fate by using N2B27 medium[11] to investigate the function of PCGF5 during neural differentiation of mESCs. To monitor the dynamic changes of gene expression during neural differentiation of mESCs upon PCGF5 loss-of-function, we performed RNA-sequencing (RNA-seq) experiments at day 0 and day 6 after neural differentiation with both wild type and *Pcgf5*$^{-/-}$ mESCs. Although loss of *Pcgf5* did not affect the pluripotency cell state, 189 genes were significantly differentially regulated (>3-fold) in *Pcgf5*$^{-/-}$ mESCs compared with control mESCs (Supplementary Data 1). In day 6 neural-differentiated mESCs, RNA-seq results indicated a more substantial change, as 579 genes were significantly upregulated and 341 genes were downregulated by PCGF5 loss-of-function at day 6 after neural differentiation (Supplementary Data 1). Further, heatmap and gene set enrichment analysis (GSEA) revealed the depletion of neural-related genes and enrichment of mESC genes in day 6 differentiated *Pcgf5*$^{-/-}$ mESCs compared with wild type (Fig. 1e, f). Gene ontology (GO) analysis revealed that the downregulated genes were enriched for neuroectoderm markers and were mainly involved in neuron differentiation and neuron development (Supplementary Fig. 2a). On the other hand, a significant enrichment of markers of stem cell maintenance and differentiation was observed in the upregulated genes (Supplementary Fig. 2b). Western blot indicated that PCGF5 loss-of-function significantly inhibited the expression of the neural markers NESTIN, β-III-TUBULIN, while the levels of the pluripotency factors OCT4 and NANOG remained high in *Pcgf5*$^{-/-}$ cells, in contrast to wild type cells (Fig. 1g), a pattern mirrored in mRNA levels, as *Nestin, Sox1* and *β-III-Tubulin* failed to robustly upregulate (Fig. 1h–j), while the levels of *Oct4 (Pou5f1)* and *Nanog* showed resistance to downregulation after PCGF5 loss-of-function (Fig. 1k–m). This suggests that PCGF5 loss-of-function inhibits mESC neural differentiation, and (at least indirectly) leads to the maintenance of mESC pluripotency.

To investigate whether ectopic expression of PCGF5 can rescue the *Pcgf5*$^{-/-}$ phenotype during neural differentiation, we established cell lines stably overexpressing PCGF5 in *Pcgf5*$^{-/-}$ mESCs and then induced neural differentiation of wild type, *Pcgf5*$^{-/-}$ and PCGF5 overexpressed in *Pcgf5*$^{-/-}$ mESCs. However, the PCGF5 overexpressing cells could only express relatively modest levels of *Pcgf5*, and missing the regulatory enhancers that the genomic context provides, *Pcgf5* was not upregulated in the same

temporal pattern as mESCs differentiate to a neural cell fate (Supplementary Fig. 3a). Nonetheless, as mESCs were induced to differentiate towards a neural cell fate, the forced expression of *Pcgf5* in the *Pcgf5⁻/⁻* mESCs could rescue the upregulation of *Nestin*, *β-III-Tubulin* and *Sox1*, and cells could again correctly downregulate *Oct4* and *Nanog* (Supplementary Fig. 3b–f). In addition, immunofluorescence staining of the neural progenitor markers NESTIN and PAX6 indicated PCGF5 overexpression could rescue the neural differentiation program (Supplementary

Fig. 3g, h). Overall, overexpression of PCGF5 could rescue the differentiation defects of *Pcgf5⁻/⁻* mESCs.

**Loss of PCGF5 activates TGF-β signaling pathway during NPC (neural precursor cell) induction**. To further explore the underlying molecular mechanisms of PCGF5-mediated neural differentiation, we performed KEGG pathway analysis on differentially expressed genes at day 6 of neural induction between

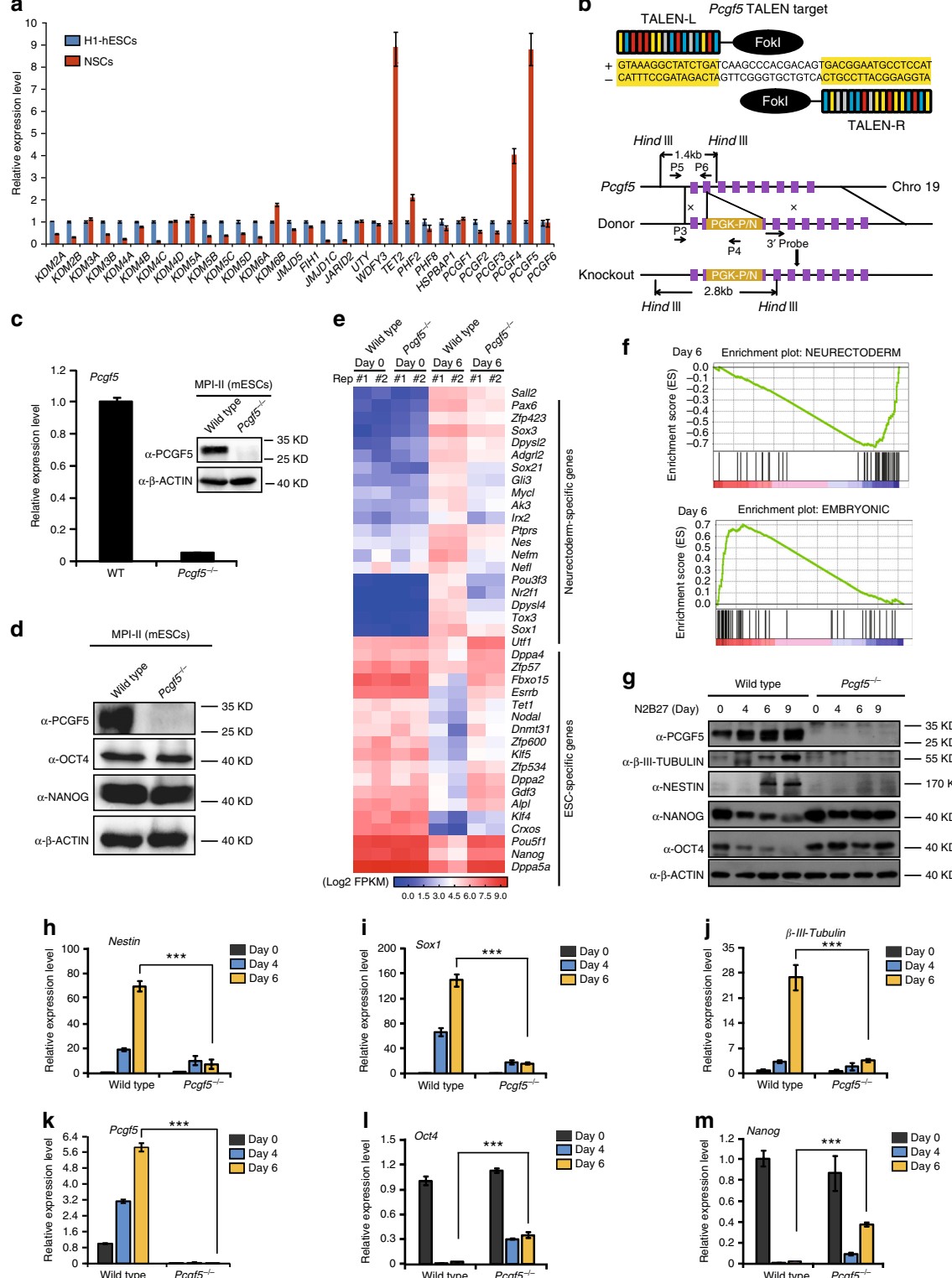

wild type and $Pcgf5^{-/-}$ mESCs. This revealed that the PCGF5-dependent genes were highly enriched for gene networks regulating TGF-β signaling pathway (Fig. 2a). Heatmap analysis suggested that PCGF5 loss-of-function at day 6 after neural differentiation significantly upregulated the expression of selected TGF-β related genes (Fig. 2b).

To explore the effect of loss-of-function of PCGF5 in neural differentiation, we performed qRT-PCR experiments which indicated that the TGF-β signaling pathway genes *Nodal*, *Lefty1* and *Lefty2* were all significantly upregulated at day 6 after induction to neural differentiation (Fig. 2c). We further asked whether SMAD2 phosphorylation (pSMAD2) (an indicator of TGF-β signaling pathway)[12] was affected by PCGF5 loss-of-function. Indeed, the pSMAD2 level was increased in differentiated $Pcgf5^{-/-}$ mESCs at day 6, which was also reversed by LY2109761 (TβRI/II inhibitor) treatment (Fig. 2d). This suggests that PCGF5 loss-of-function at least partially affects neural differentiation by upregulation of the TGF-β signaling pathway. To investigate whether inhibition of TGF-β signaling pathway can rescue the phenotype of neural differentiation, we performed immunofluorescence staining to examine the distribution of NESTIN and PAX6 in wild type and $Pcgf5^{-/-}$ mESCs with or without LY2109761 treatment. Our data also showed that treatment of $Pcgf5^{-/-}$ mESCs with LY2109761 also partially rescued the expression of NESTIN and PAX6 caused by PCGF5 loss-of-function (Fig. 2e, f).

To further verify if TGF-β signaling pathway is involved in suppression of neural differentiation by PCGF5 loss-of-function, we took advantage of an mESC lines that expresses GFP under the control of the *Sox1* promoter (Sox1-GFP mESCs)[13], which is one of the earliest neuroectodermal markers induced in the embryo[14]. As in normal mESCs, when we knocked down PCGF5, the expression of NANOG and OCT4 was unaffected (Supplementary Fig. 4). Next, we differentiated cells towards a neural cell fate[15] using wild type, PCGF5-deficient mESCs treated with either LY2109761, SB431542 (TGF-β signaling: ALK4/5/7 inhibitor) or LDN-193189 (BMP signaling: ALK2/3 inhibitor). qRT-PCR analysis indicated that the expression of the NPC marker *Sox1* decreased significantly after PCGF5 loss-of-function; however, treatment of PCGF5-deficient cells with LY2109761 for 6 days after neural differentiation restored the expression of *Sox1* compared with treatment of PCGF5-deficient cells with DMSO (Fig. 2g). Flow cytometry analysis indicated that PCGF5 loss-of-function in two mESC clones reduced the percentage of Sox1-GFP$^+$ cells (32 and 44%) compared with wild type mESCs (74 and 79%; Fig. 2h, i). Interestingly, treatment of PCGF5-deficient mESCs with LY2109761 or SB431542 could restore the expression of the Sox1-GFP reporter at day 6 after neural differentiation, from 32.8 to 62.3, 44.3 to 73.7% for LY2109761 treatment, and from 32.8 to 51.1, 44.3 to 58.7% for SB431542 treatment (Fig. 2h, i). However, treatment of $Pcgf5^{-/-}$ mESCs with LDN-193189 had little effect on the fraction of Sox1-GFP positive cells, from 32.8 to 33.6%, 44.3 to 46.8% for the two PCGF5-deficient cell lines, at day

6 after neural differentiation compared with control treatment (Fig. 2h, i). Taken together, these results suggest that PCGF5-mediated regulation of the TGF-β signaling pathway is a critical pathway mediating neural fate determination in mESCs.

**PCGF5 stimulates RING1B-dependent histone H2AK119ub1.** To determine whether PCGF5 interacts with histone H2A, we transfected either Flag or Flag-PCGF5 into 293T cells and performed Flag co-IP experiments. We found that Flag-PCGF5 could bind to histone H2A (Fig. 3a). Moreover, Flag co-IP experiments indicated that RING1B was retained by Flag-PCGF5, but not by Flag alone (Supplementary Fig. 5a). To examine which domain of PCGF5 interacts with RING1B, we performed Flag co-IP experiments with either Flag-PCGF5 or Flag-PCGF5 without a ring finger domain and found that RING1B no longer bound to PCGF5 once the ring finger was deleted (Fig. 3b, c), suggesting that the ring finger domain of PCGF5 is required for this interaction. To determine the domains of RING1B responsible for binding to PCGF5, we expressed and purified fusion proteins for GST-RING1B and a series of deletions (Fig. 3d) in *BL21* cells, then carried out GST pull-down assays. The results indicated that the ring finger and C-terminal domains of RING1B (amino acids 49–95 and amino acids 94–336, respectively) bound to PCGF5 (Fig. 3e). Flag co-IP and sucrose gradient experiments showed that PCGF5 formed a complex with RYBP but not CBX7 (Fig. 3f, g), suggesting PCGF5 belongs to a noncanonical PRC1 complex[4,5].

Since PCGF5 contains a ring finger domain and the domain has been identified as a signature motif for ubiquitin E3 ligase[16], we were interested to know the role of PCGF5 in regulating RING1B-dependent histone H2A ubiquitylation. To answer this question, we established an in vitro ubiquitination assay by incubating nucleosome, E1/E2 ligases, HA tagged ubiquitin and different amounts of purified GST-RING1B, GST-PCGF5 or GST-PCGF4 fusion proteins. Interestingly, we found that induction of RING1B-dependent histone H2A ubiquitylation at K119 by PCGF5 was in a dose-dependent manner (Fig. 3h, lane 3–5). Further, we found that GST-PCGF4 fusion protein alone could not efficiently promote histone ubiquitination (Fig. 3h, lane 6). But, similar to PCGF5, once RING1B fusion protein was added to the reaction, histones were efficiently ubiquitinated (Fig. 3h). These results suggest that in vitro purified PCGF5 fusion protein does not have E3 ligase activity, but can stimulate RING1B-dependent histone H2A ubiquitylation. Our data further indicated that ring finger domain of PCGF5 was required for PCGF5-mediated RING1B-dependent histone H2A ubiquitylation (Fig. 3i). To investigate if PCGF5 promotes histone H2A ubiquitylation in vivo, we transfected Flag-PCGF5 into 293T cells and performed immunofluorescence staining experiments. Our data indicated that Flag-PCGF5 overexpression significantly increased the level of histone H2A ubiquitylation compared with cells without Flag-PCGF5 overexpression (Supplementary Fig. 5b–d).

**Fig. 1** PCGF5 loss-of-function blocks mESC neural differentiation. **a** Gene expression analysis of epigenetic factors in human embryonic stem cells (H1) and human neural stem cells (NSCs), respectively ($n = 3$). Results are shown relative to H1. **b** Schematic overview depicting the targeting strategy for the *Pcgf5* locus using TALENs (PGK/PN: donor indicates that containing a loxP-flanked PGK-puromycin cassette and loxp-flanked PGK-neomycin cassette. PGK human phophoglycerol kinase promoter, P puromycin resistance gene, N neomycin resistance gene). **c** Western blot and qRT-PCR analysis of PCGF5 expression in wild type and $Pcgf5^{-/-}$ mESCs. Results are shown relative to wild type ($n = 3$). **d** Western blot analysis of PCGF5, NANOG and OCT4 expression in wild type and $Pcgf5^{-/-}$ mESCs. **e** Heatmap illustrating the expression of selected neurectoderm genes and mESC-specific genes that were shown as log2 FPKM in wild type and $Pcgf5^{-/-}$ mESCs at day 6 after neural differentiation. Each lane corresponds to an independent biological sample. **f** GSEA profiles of the sets of neurectoderm genes and mESC-specific genes. **g** Western blot analysis of PCGF5, pluripotent markers (OCT4, NANOG), neural markers (NESTIN, β-III-TUBULIN) in wild type and $Pcgf5^{-/-}$ mESCs during neural differentiation of mESCs. **h–m** Gene expression analysis of *Nestin*, *Sox1*, *β-III-tubulin*, *Pcgf5*, *Oct4*, *Nanog*, in wild type and $Pcgf5^{-/-}$ mESCs during neural differentiation of mESCs ($n = 3$). Results are shown relative to wild type at day 0. Data in **a**, **c**, **h–m** are represented as mean values ± s.d. with the indicated significance from Student's *t*-test (***$p < 0.001$)

**PCGF5 is targeted to TGF-β target genes during NPC induction**. To investigate PCGF5 downstream targets at a genome-wide scale during neural differentiation of mESCs, we attempted to perform chromatin immunoprecipitation (ChIP) followed by deep sequencing (ChIP-seq) for PCGF5 by using commercial antibodies. Unfortunately, these commercial anti-PCGF5 antibodies did not work for ChIP experiments. Therefore, we generated Flag-tagged PCGF5 knockin cell lines which stably express PCGF5-Flag in the Sox1-GFP mESCs (Fig. 4a and Supplementary

Fig. 6a). Knockin of PCGF5 in mESCs had no effect on pluripotency gene expression (*Oct4, Nanog, Klf4*) (Supplementary Fig. 6b). PCGF5 loss-of-function did not affect the maintenance of stem feature of mESCs, however, loss of PCGF5 prevented mESC neural differentiation. Therefore, to identify PCGF5 downstream targets during neural differentiation, we performed ChIP-seq by using an anti-FLAG antibody in NPCs at day 6 after neural differentiation of mESCs (Fig. 4b, c). Our ChIP-seq data identified 12,015 peaks in NPCs for PCGF5 (Supplementary

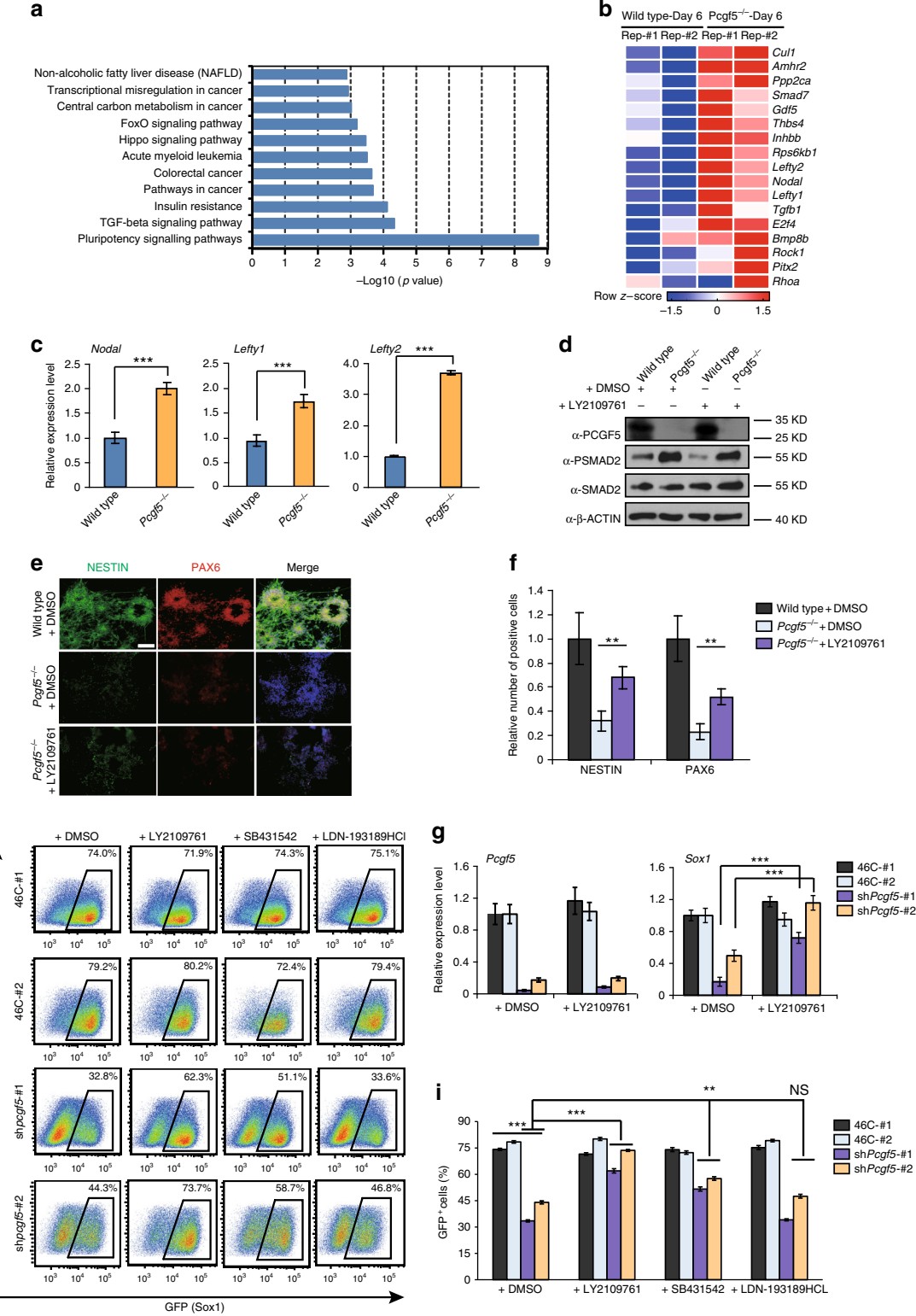

Data 2). ChIP-seq analysis revealed a preferential distribution of PCGF5 near transcription start sites (TSS) of genes in NPCs (Fig. 4d). About 25.04% of PCGF5 sites are near promoter regions. 35.87% of the PCGF5-binding sites are located in the intergenic regions, a significant number of PCGF5-binding sites fall within genes, with 33.55% in the introns and 5.54% in the exons (Fig. 4e). We further focused on the effects of PCGF5 on the genes in NPCs. Among 80 genes in the TGF-β signaling pathway, 35 (43.8%) of them were directly targeted by PCGF5 (Supplementary Data 2). Combination analysis of PCGF5 ChIP-seq and RNA-seq data at day 6 after neural differentiation in both wild type and PCGF5-deficient cells indicated that 521 PCGF5-bound genes were downregulated, while 99 PCGF5 target genes were upregulated by the loss of PCGF5 (Fig. 4f). GO analysis indicated that these downregulated genes were associated with axon generation of neurons, neuron differentiation (Fig. 4g), further suggesting that PCGF5 might be required for the activation of NPC-related genes. And the upregulated genes targeted by PCGF5 were involved in cell proliferation, cellular response to endogenous stimulus (Fig. 4g and Supplementary Data 2). Included within these 99 upregulated PCGF5 target genes were *Nodal*, *Lefty1* and *Lefty2*, three TGF-β signaling pathway genes. Hence, we performed ChIP-qPCR experiments to examine the recruitment of PCGF5 onto the promoters of these genes. Our data indicated that PCGF5 was indeed specifically recruited to the promoters of *Nodal*, *Lefty1* and *Lefty2* at day 6 of neural differentiation but not in mESCs (Fig. 4h).

PRC1-dependent H2AK119ub1 leads to the recruitment of PRC2 and H3K27me3 to effectively initiate a polycomb domain[7], to investigate PCGF5 loss-of-function on the deposition of H2AK119ub1 and H3K27me3, we performed ChIP-qPCR with anti-H2AK119ub1 and anti-H3K27me3 antibodies in both wild type and PCGF5-deficient mESCs and NPCs (at day 6 of neural differentiation). Our data indicated that, although PCGF5 loss-of-function led to reduction of H2AK119ub1 at *Lefty1* and *Lefty2* promoters, there were no changes at the *Nodal* promoter in mESCs, PCGF5 loss-of-function significantly reduced the level of H2AK119ub1 at the promoters of these genes at day 6 of neural differentiation (Fig. 4i). Furthermore, loss of PCGF5 had no effects on the level of H3K27me3 at the promoters of *Nodal*, *Lefty1* and *Lefty2* in mESCs, but significantly decreased the level of H3K27me3 at the promoters of these genes in NPCs (Fig. 4j), which is consistent with de-repression of these genes by loss of PCGF5 during neural differentiation.

**Loss of PCGF5 affects H2AK119ub1 and H3K27me3 distributions**. To investigate if PCGF5 regulates the genome-wide distribution of histone H2AK119ub1 and H3K27me3 during neural differentiation, we further performed ChIP-seq by using anti-H2AK119ub1 and H3K27me3 antibodies in both wild type and PCGF5-deficient mESCs and NPCs (at day 6 of neural differentiation). Western blot indicated that histone H2AK119ub1

protein level remained unchanged but histone H3K27me3 protein level was gradually decreased during neural differentiation in both wild type and PCGF5-deficient cells (Supplementary Fig. 7a, b). Although loss of PCGF5 had little change on the genome-wide distributions of both H2AK119ub1 and H3K27me3 (Fig. 5a, b and Supplementary Fig. 7c, d), loss of PCGF5 significantly changed the enrichments of both H2AK119ub1 and H3K27me3 at day 6 of neural differentiation (Fig. 5c, d). We observed many common H2AK119ub1 ($n = 1425$ peaks) and H3K27me3 peaks ($n = 5081$ peaks) in both mESCs and NPCs. There are also large groups of peaks that were only enriched in mESCs for H2AK119ub1 ($n = 7247$ peaks) and H3K27me3 ($n = 7683$ peaks) or NPCs for H2AK119ub1 ($n = 4378$ peaks) and H3K27me3 ($n = 6727$ peaks) (Fig. 5c, d). PCGF5 loss-of-function significantly inhibited the reduction of ESCs-enriched peaks for both H2AK119ub1 and H3K27me3 in NPCs, respectively. In addition, loss of PCGF5 resulted in lower levels of NPCs-enriched peaks for both H2AK119ub1 and H3K27me3 in NPCs (Fig. 5c, d). We observed that PCGF5 had more peaks in NPCs than in mESCs (Supplementary Data 2), consistent with the requirement of PCGF5 for neural differentiation of mESCs (Fig. 5c, d). GO-term analysis of H2AK119ub1-associated and H3K27me3-associated genes revealed that mESC-enriched peaks were related to nervous system development and neurogenesis (Fig. 5e, f and Supplementary Data 3).

Consistent with ChIP-qPCR data, examination of ChIP-seq data at the *Nodal* locus confirmed that PCGF5 loss-of-function caused significant decreases of H2AK119ub1 and H3K27me3 near the promoter of *Nodal* at day 6 of neural differentiation (Supplementary Fig. 8). Conversely, PCGF5 loss-of-function maintained higher level of both H2AK119ub1 and H3K27me3 near the promoters of neural specific genes such as, *Sox1*, *Nestin*, *Cdh2*, *Pou3f2* (Fig. 5g), which are consistent with the neural differentiation defect of mESCs after PCGF5 loss-of-function. Further, ChIP-qPCR data indicated that PCGF5 was highly enriched at the promoter regions of *Sox1*, *Nestin*, *Cdh2*, *Pou3f2* in NPCs compared with mESCs (Fig. 5h). During mESC neural differentiation, both H2AK119ub1 and H3K27me3 at the promoter regions were dramatically reduced, however, PCGF5 loss-of-function significantly blocked the reduction of both H2AK119ub1 and H3K27me3 (Fig. 5i, j), which agrees with the silenced expression of these neural-specific genes. Overall, these data suggest that PCGF5 acts to maintain the level of H2AK119ub1 and H3K27me3 at the genes of TGF-β signaling pathway, and that this is required for the repression of these genes.

**Discussion**
As part of different PRC1 variants, the posterior sex combs (PSC) homologs participate in gene repression by enhancing ubiquitination of histone H2A[17], and by altering the molecular function of PRC1. Six *Pcgf* genes appear to be critical factors that give developmental context-specific activity to PRC1. For example,

**Fig. 2** Loss of PCGF5 activates TGF-β signaling pathway during NPC induction. **a** RNA-seq-based ingenuity pathway analysis of wild type and *Pcgf5*⁻/⁻ mESCs at day 6 of neural differentiation. Results are showed as –log10 (*p*-value). **b** Heatmap illustrating the expression changes of the selected TGF-β-related genes that were shown as row *z*-score in wild type and *Pcgf5*⁻/⁻ mESCs at day 6 after neural differentiation. Each lane corresponds to an independent biological sample. **c** Gene expression analysis of the *Nodal, Lefty1, Lefty2* in wild type and *Pcgf5*⁻/⁻ mESCs at day 6 after neural differentiation ($n = 3$). Results are shown relative to wild type at day 6. **d** Western blot analysis of pSMAD2 in wild type and *Pcgf5*⁻/⁻ cells at day 6 after neural differentiation. DMSO or LY2109761 (1 μM) was added during neural differentiation. **e** Immunostaining of the neural progenitor markers NESTIN and PAX6 in wild type and *Pcgf5*⁻/⁻ mESCs at day 6 after neural differentiation. Scale bar represents 100 μm. **f** Statistical analysis of positive cells expressing NESTIN or PAX6 described in **e** ($n = 3$). **g** Gene expression analysis of *Pcgf5* and *Sox1* at day 6 after neural differentiation. DMSO, LY2109761 (1 μM) was added into the media during neural differentiation ($n = 3$). Results are shown relative to wild type at day 6 after neural differentiation. **h** Summary of FACS data from Sox1-GFP expression in control or PCGF5-deficient mESCs at day 6 after neural differentiation. The cells were treated with DMSO, LY2109761 (1 μM), SB431542 (1 μM) or LDN-193189 HCl (100 nM) during neural differentiation ($n = 3$). **i** Statistical analysis of positive cells expressing Sox1-GFP described in **h**. Data in **c**, **f**, **g**, **i**, are represented as mean values ± s.d. with the indicated significance from Student's *t*-test (NS, no significant, **$p < 0.01$, ***$p < 0.001$)

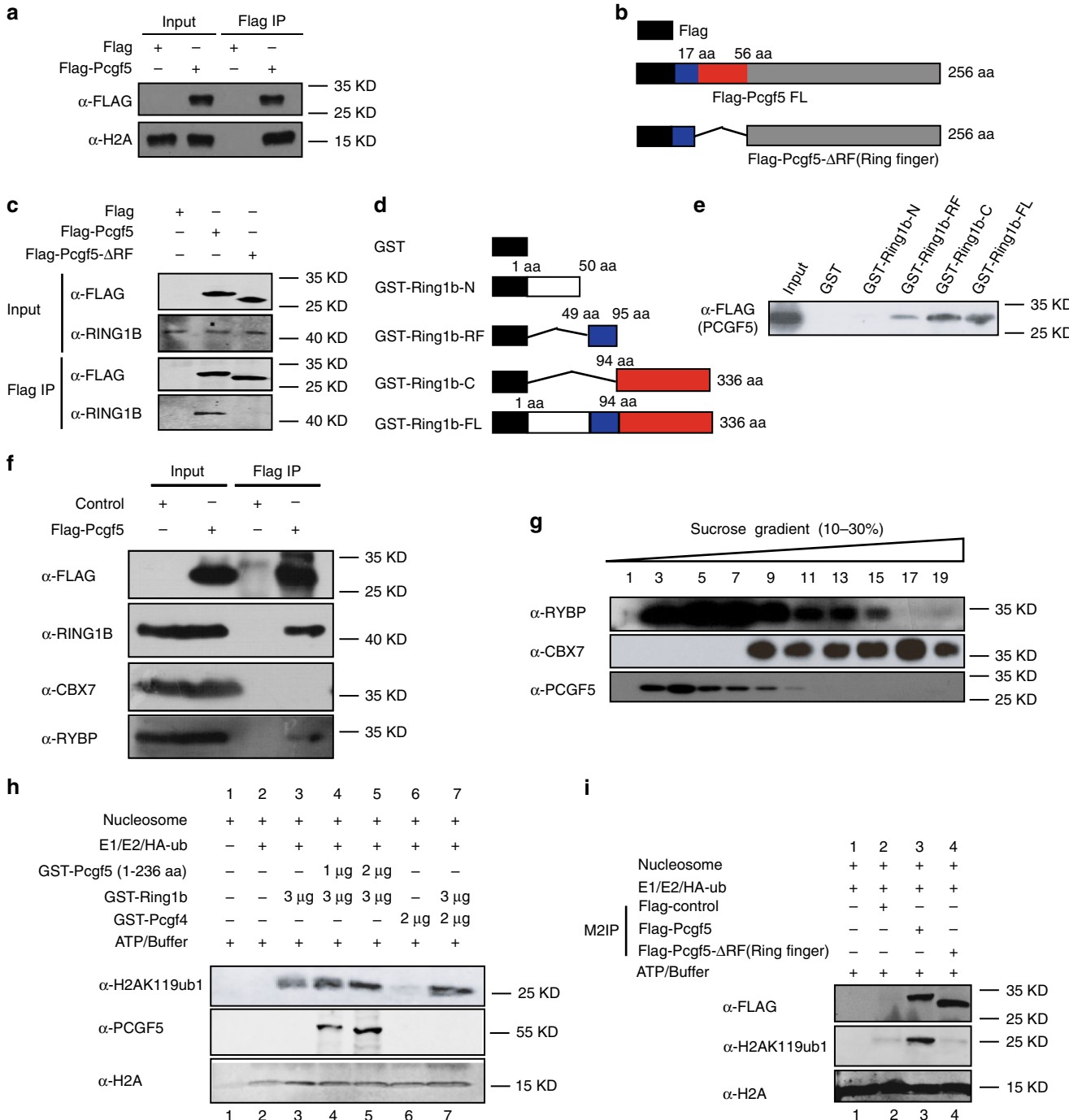

**Fig. 3** PCGF5 stimulates RING1B-dependent histone H2AK119ub1. **a** Detection of the interaction between Flag-PCGF5 and H2A. Flag-tagged empty vector was used as control. **b** Schematic representation of Flag-PCGF5 and its deletions used in Flag co-IP assays. **c** Detection of the interaction between RING1B and PCGF5 with or without a Ring-finger. **d** Schematic representation of GST-RING1B and its deletions used in GST pull-down assays. **e** Proteins probed with anti-FLAG antibody by Western blot. The beads were incubated with cell lysates from the Flag-PCGF5 overexpressed in 293T cells. **f** Detection of the interaction between Flag-PCGF5 and PRC1 subunits by Flag co-IP in 293T cells. Flag-tagged empty vector was used as a control. **g** Sucrose gradient analysis of whole cell extracts of mESCs. Every fraction from a 10–30% sucrose gradient was further resolved on SDS-PAGE followed by immunoblotting for the indicated antibodies. **h** In vitro H2A monoubiquitinylation assays with nucleosome, GST purified PCGF5, PCGF4, RING1B, E1, E2 and ubiquitin. The reactions were resolved on SDS–PAGE followed by immunoblotting. **i** In vitro H2A monoubiquitinylation assays with purified Flag-tagged PCGF5 and Flag-tagged PCGF5 without a Ring-finger in 293T cells. The reactions were resolved on SDS-PAGE followed by immunoblotting

PCGF1 (NSPC1), together with the transcription factor RUNX1, regulate HSC differentiation and self-renewal[18]. PCGF2 (MEL-18) positively regulates the expression of key mesoderm transcription factors, and is involved in gene activation during cardiac differentiation[19]. *Pcgf3/5* gene knockout results in female-specific

embryo lethality and abrogates Xist-mediated gene repression[20]. Previous reports showed that in the absence of *Pcgf6*, its target genes had specific losses of H2AK119ub1 while other PRC1 target genes did not lose any H2AK119ub1[17,21]. Similarly, recent observations have shown that the global level of H2AK119ub1 is

unchanged in $Pcgf1^{-/-}$ ES cells[22]. These alterations in molecular activity of PRC1 lead to changes in developmental outcome.

Here we show that PCGF5 is highly expressed in NPCs compared with ESCs and appears to be a major factor regulating neural differentiation of mESCs. RNA-seq experiments demonstrated that PCGF5-deficient mESCs were not capable of fully

differentiating towards the neural lineage, and pluripotency-specific genes, which are normally expressed in mESCs but silenced during neural differentiation, were not properly switched off. PCGF5 has been demonstrated to contribute to H2AK119ub1-dependent recruitment of PRC2 and H3K27me3 modification in a manner similar to other non-canonical PRC1 complexes in

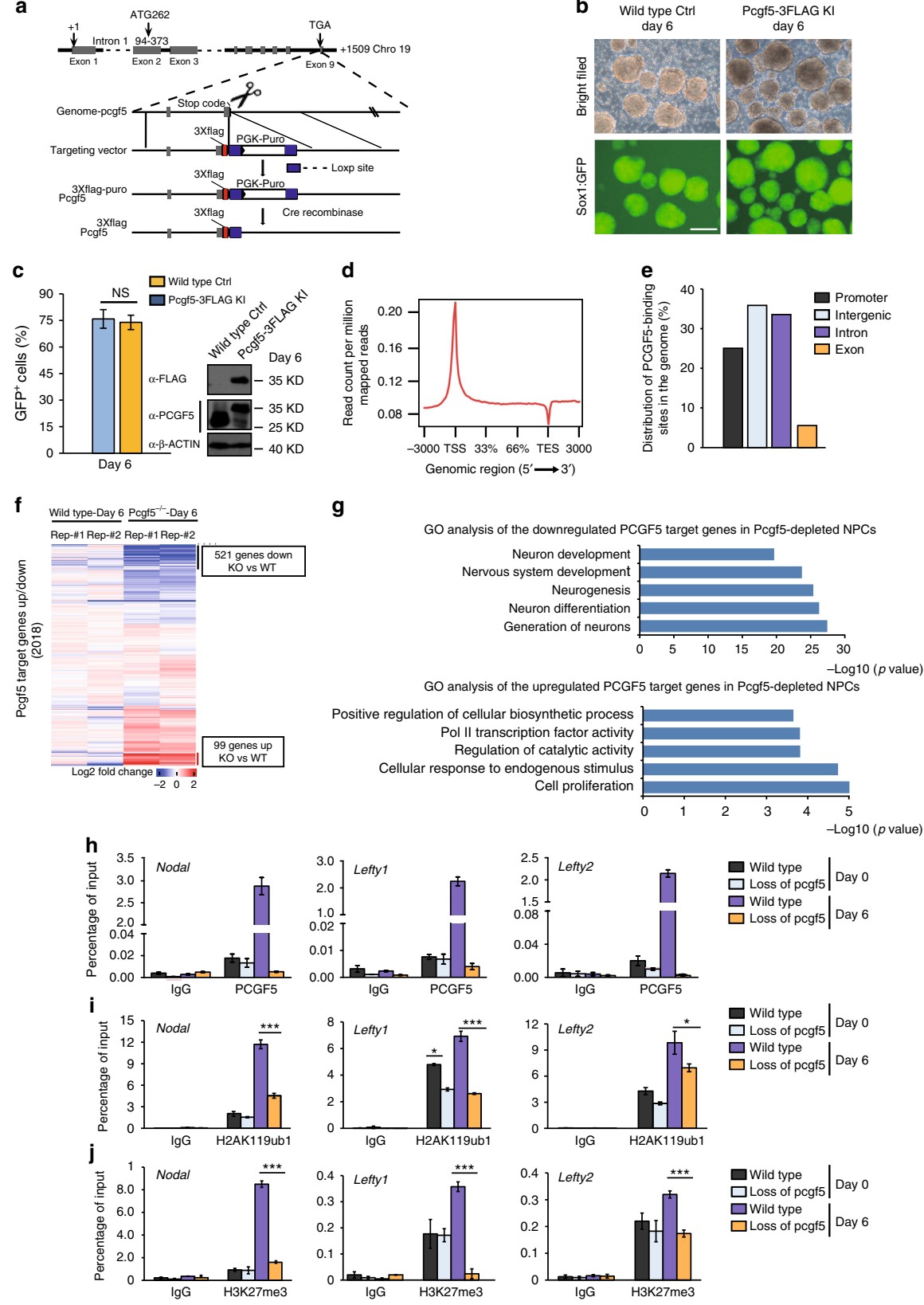

mESCs[7]. PCGF5 loss-of-function clearly showed reduction in H2AK119ub1 level in hematopoietic stem and progenitor cells[23].

Previous reports revealed that PCGF5-PRC1-AUTS2 complex is involved in gene activation by transcriptional co-activator P300 after RING1B is phosphorylated by the CK2 protein[8]. We investigated whether PCGF5 co-binds to specific targets with histone repressive marks, H2AK119ub1, H3K27me3 and active marks H3K27ac[24] at day 6 after neural differentiation. Our data indicated that only a small fraction of PCGF5 target genes overlapped with the repressive H2AK119ub1-enriched and H3K27me3-enriched genes (Supplementary Fig. 9a). However, we surprisingly found that most PCGF5 target genes overlapped with the genome-wide localizations of active marks H3K27ac and H3K4me3[24] in NPCs (Supplementary Fig. 9b). Our data further indicated that about 39.5% (1024) of PCGF5-binding sites target to the active genes and only 6.5% (170) of PCGF5-binding sites target silent genes (Supplementary Fig. 9c). These data suggest that PCGF5 may function as an activator of NPC-specific genes during mESC neural differentiation.

Our data suggest that PCGF5 not only acts to maintain the levels of H2AK119ub1 and H3K27me3 at the key genes of TGF-β signaling pathway, and that this is required for the repression of these genes, but also facilitates the reduction of H2AK119ub1 and H3K27me3 around the promoters of neural-specific genes during neural differentiation, suggesting PCGF5 plays dual functions in regulating mESC neural differentiation, acting as a repressor for TGF-β signaling pathway and functioning as an facilitator for neural-related genes in another unknown mechanism.

In summary, this study highlights how PCGF5 acts to bring context specificity to the functions of PRC1 in controlling lineage-specific gene expression and cell fate determination.

## Methods

**Cell culture and differentiation.** hESCs lines H1 (Wi Cell) were maintained in mTeSR1 (STEMCELL Technologies) on matrigel (Corning)-coated plates. mESC lines MPI-II was kindly provided by Dr. Jing Liu. Sox1-GFP mESC lines (a kind gift from Dr. Naihe Jing)[13] were cultured on mitomycin C-inactivated mouse embryo fibroblasts in Dulbecco's modified Eagle medium (DMEM) high-glucose media containing 15% fetal bovine serum (FBS; Gibco), 1 mM sodium pyruvate (Gibco), 1 mM non-essential amino acids (Gibco), 1× GlutaMAX (Gibco), 0.1 mM 2-mercaptoethanol (Gibco), 1000 U/ml leukemia inhibitory factor, and the 2i inhibitors (3 μM CHIR99021, and 1 μM PD0325901). mESCs stably expressing PCGF5 shRNA (Supplementary Table 1) were generated by infection with lentiviral pLKO.1 vectors and then selected with puromycin.

To induce neural differentiation, monolayer culture for neural differentiation was performed[13]. Briefly, cells were dissociated and plated at a density of $1 \times 10^4$ cells/cm$^2$ in N2B27 medium supplemented with 1 mM L-glutamine and 0.1 mM β-mercaptoethanol. For neural differentiation in suspension[15], the dissociated mESCs were suspended in the Petri dish and cultured in GMEM (Gibco) supplemented with 8% knockout serum replacement (Gibco), 1 mM Glutamine, 1 mM pyruvate, 0.1 mM nonessential amino acids and 0.1 mM b-mercaptoethanol (Gibco). To initiate hESCs neural differentiation, hESCs were plated onto matrigel-coated plates 12-well plates with 80–100% of cell confluence, and then these cells were cultured in N2B27 medium plus SB431542 and Dorsomorphin (50% DMEM/F12

(Hyclone), 50% Neurobasal (Gibco), N2 (Gibco, 200×), Glutamax (Gibco, 200×), NEAA (Gibco, 200×), 1 μg mL-1 heparin (Sigma), 5 μg mL$^{-1}$ insulin (Gibco, 200×), 5 μM SB431542 (Selleck), 5 μM Dorsomorphin (DM, Selleck). Fresh culture medium was changed every 2 days. After 8 days, the cells were passaged on matrigel-coated six-well plates in N2B27 medium.

**Generation of PCGF5 knockout mESCs.** To generating TALEN-mediated PCGF5 knockout in mESCs, left and right homology arms were amplified from genomic DNA for donor DNA. A loxP-flanked PGK-puromycin cassette or loxP-flanked PGK-neomycin cassette were cloned between two homology arms in the pUC-57 vector using primers p1 and p2. For targeting, $1 \times 10^6$ mESCs were electroporated with 1 μg of donor DNA and 2 μg of each TALEN plasmid. Then the cells were plated onto feeder-coated tissue culture plates for 1 day. Positive clones were selected by puromycin (1 μg/ml) or G418 (200 μg/ml)[25]. The selected colonies were verified by genomic PCR and Southern blot.

**PCR verification of corrected clones.** PCR was performed using LA Taq (Takara) according to the manufacturer's instructions. 50–100 ng of genomic DNA templates were used in all reactions. Primers including P3 (upstream of 5′ homology arm) and P4 (in the drug resistance cassette) were used to amplify a 1.0-kb product of the 5′ junction of a targeted integration. Primers including P5 (on 5′ homology arm) and P6 (on 3′ homology arm) were used to amplify a 2.5-kb product or a 1.0-kb product to identify whether random integration occurred (Fig. 1b). All of the primers used are listed in Supplementary Table 1.

**Generation of PCGF5-deficient stable 46 C mESC lines.** We used the RNAi lentivirus system for generation of PCGF5-deficient stable 46 C mESC lines. In brief, shRNA sequence-targeting mouse PCGF5 was cloned into pLKO.1. The recombinant construct, as well as two assistant vectors psPAX2 and MD2.G, were co-transfected into HEK293T cells. Viral supernatants were collected 48 h later, filtrated through 0.45 μm filters. The viruses were used to infect $5 \times 10^6$ mESCs in a 6-cm dish with 8 μg/ml polybrene. The clonal cell clusters of mESCs were picked out with micro-needle after 2 μg/ml puromycin (Amresco) selection.

**Generation of 3× Flag tagged PCGF5 mESC stable cell lines.** The sgRNA target sequence was inserted into the plasmid pX330. Then, pX330 along with the linearized PCGF5 3× Flag targeting vector were electroporated into mESCs for gene editing. The correctly targeted colonies were chosen through drug selection, PCR verification and genomic DNA sequencing. After that, the drug-resistant gene was removed with Cre recombinase to obtain the final targeted 3× Flag tagged PCGF5 mESCs, which was further verified by Western blot. The sgRNA sequences used are listed in Supplementary Table 1.

**Southern blot.** To detect homologous recombination at Pcgf5 locus, an 852-bp Pcgf5 specific probe in the 5′ side of the left homology arm was synthesized by PCR amplification and labeled with a DIG-dUTP labeling kit (Roche Applied Science). Genomic DNA was digested with Xho I and Nde I, and then standard Southern blot was performed following the instruction manuals of DIG High Prime DNA labeling and detection starter kit II (Roche Applied Science).

**Cell immunofluorescence.** Cells were fixed in 4% paraformaldehyde and permeabilized with 0.2% Triton X-100 containing 10% FBS (Invitrogen)/1% BSA in PBS at room temperature for 15 min. Samples were then incubated with primary antibodies overnight at 4 °C. The antibodies used for cell immunofluorescence were against OCT4, NANOG, NESTIN, PAX6, FLAG and H2AK119ub1, respectively. The cells were then washed for four times and 0.1 μg/ml DAPI (Sigma) was included in the final wash to stain nuclei. Images were captured with an inverted microscope (DMI4000, Leica Microsystems).

**Fig. 4** Analysis of PCGF5 binding sites in NPCs and the PCGF5-driven deposition of H2AK119ub1 and H3K27me3 at TGF-β-associated genes during neural differentiation. **a** Strategy of generating Flag-tagged PCGF5 knockin stable cell lines in Sox1-GFP knockin mESCs. **b** Representative cellular morphologies at day 6 of neural differentiation ($n = 3$). The dissociated wild type and Pcgf5-3 × Flag knockin mESCs were suspended in the Petri dish and differentiated in KSR medium for 6 days. Scale bar represents 100 μm. **c**. Statistical analysis of Sox1-GFP cells at day 6 of neural differentiation and Western blot analysis of the Flag (Flag-PCGF5), PCGF5 in wild type and Pcgf5-3 × Flag knockin mESCs at day 6 after neural differentiation. **d** The tag density heatmap plot for PCGF5 binding signal. **e** Genome-wide distribution of PCGF5 binding sites in NPCs at day 6 after neural differentiation. **f** Heat map illustrating the expression changes of PCGF5 target genes that were shown as log2 fold change in wild type and Pcgf5$^{-/-}$ mESCs at day 6 after neural differentiation. Each lane corresponds to an independent biological sample. **g** GO analysis of biological functions of upregulated and downregulated PCGF5 target genes in wild type and Pcgf5$^{-/-}$ mESCs at day 6 of neural differentiation. Results are expressed as –log10 (p-value). **h** ChIP-qPCR analysis of PCGF5 occupancies at promoter regions of Nodal, Lefty1 and Lefty2 in both wild type and PCGF5-deficient mESCs and NPCs (at day 6 of neural differentiation) ($n = 3$). **i** ChIP-qPCR analysis of H2AK119ub1 occupancies at promoter regions of Nodal, Lefty1 and Lefty2 in both wild type and PCGF5-deficient mESCs and NPCs (at day 6 of neural differentiation) ($n = 3$). **j** ChIP-qPCR analysis of H3K27me3 occupancies at promoter regions of Nodal, Lefty1 and Lefty2 in both wild type and PCGF5-deficient mESCs and NPCs (at day 6 of neural differentiation) ($n = 3$). Data in **c**, **h–j** are represented as mean values ± s.d. with the indicated significance from Student's t-test (NS no significant, *p < 0.05, ***p < 0.001)

**Quantitative RT-PCR (qRT-PCR).** Total RNA was isolated from samples with Trizol reagents (Invitrogen). Any potential DNA contamination was removed by RNase-free DNase treatment (Promega). One microgram of total RNA was then reverse-transcribed with Superscript First-Strand Synthesis system (Promega). cDNAs of interest were then quantified with real-time qPCR amplification. The primers used in the qRT-PCR assays are listed in Supplementary Table 2. All the experiments were repeated for three times.

**In vitro ubiquitination assay.** In vitro ubiquitin ligase reaction[26] was performed by incubating 1 μl of ubiquitin activating enzyme E1 (Boston Biochem, 100 nM), 1 μl of ubiquitin conjugating enzyme UbcH5c (E2) (Boston Biochem, 500 nM), 5 μl

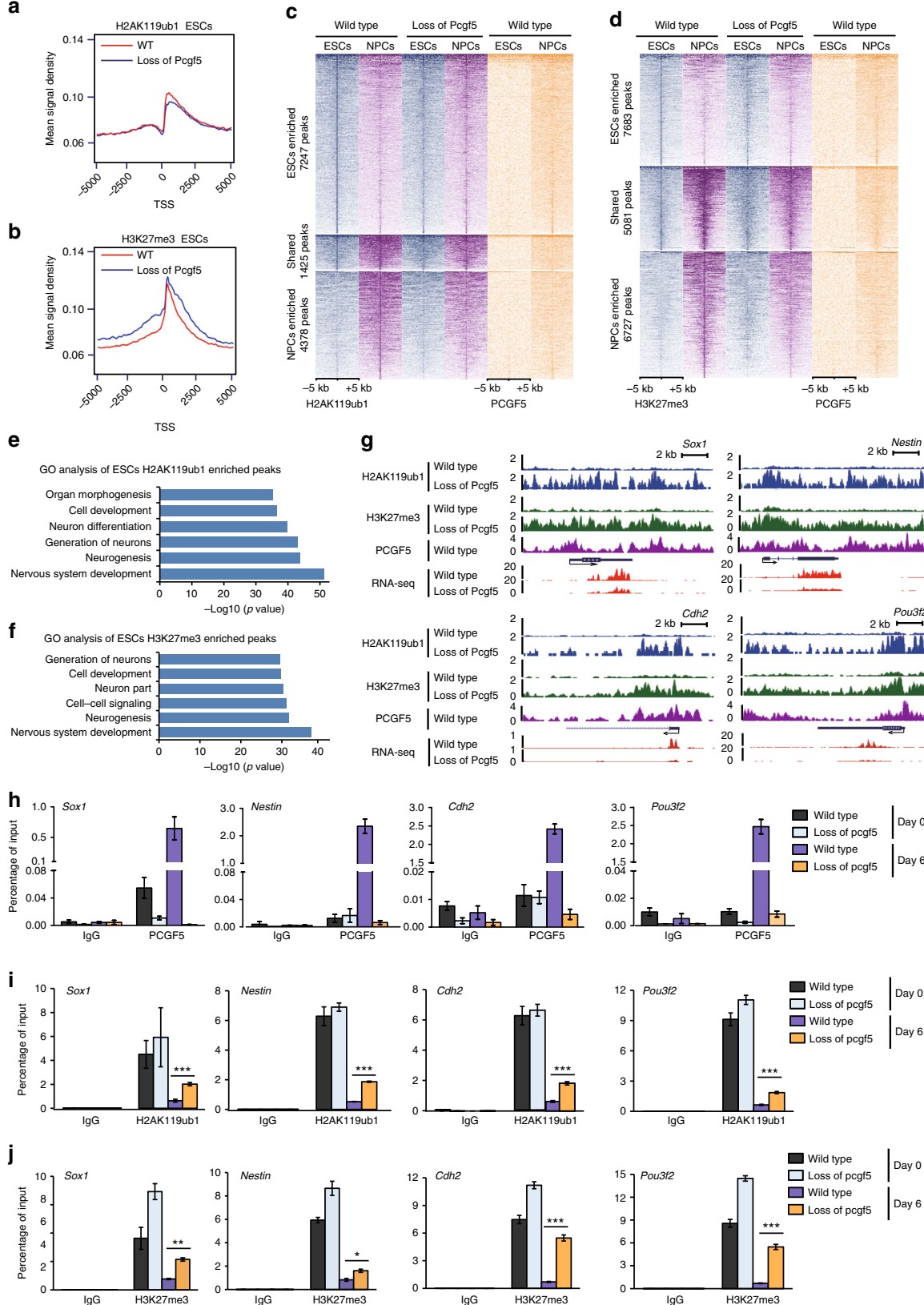

of HA-ubiquitin (Boston Biochem, 10 mM), 5 µg prepared nucleosomes with indicated GST recombinant proteins in a 25 µl reaction containing 5 mM ATP and 10× in vitro ubiquitination buffer (500 mM Tris-HCl (pH7.5), 50 mM MgCl₂, 10 mM DTT). After incubation at 37 °C for 1 h, reaction was terminated by addition of sample loading buffer. The proteins were resolved in 12% SDS–PAGE and blotted with anti-histone H2AK119ub1 antibody.

**Protein extraction and western blot**. Whole cell extracts were obtained with cell lysis buffer (50 mM Tris-HCl (pH 7.6), 1% Triton X-100, 1 mM EDTA, 10% glycerol, 1 mM DTT, 1 mM PMSF and protease inhibitor cocktail). After centrifugation at 18,000×g for 10 min, soluble proteins were quantified by BCA, loaded onto SDS-PAGE and transferred to PVDF membrane (Invitrogen). Then the membrane was washed with TBS-T buffer and Western blot was performed with the indicated antibodies. All uncropped images from Western blot can be found in Supplementary Figs.10–12. Antibodies used for Western blot are listed in Supplementary Table 3.

**Histone extraction**. Approximately $5 \times 10^6$ cells were collected and used to extract histones. Briefly, cells were washed once with 5 ml ice-cold PBS and re-suspended in 2 ml of hypotonic lysis buffer (10 mM Tris-HCl (pH 7.4), 1 mM KCl, 1.5 mM MgCl₂, 1 mM DTT, protease inhibitors), incubated on ice for 30 min. Then, the nucleus was pelleted by spinning at 2500×g at 4 °C for 10 min, resuspended in 0.4 ml H₂SO₄ (0.4 N) and rotated at 4 °C for 30 min overnight. After centrifugation at full speed for 10 min, the supernatant was precipitated with 20% TCA and incubated on ice for 30 min. Pellets were collected and washed twice with cold acetone. Pellets were resuspended in 30–50 µl of TE buffer and analyzed by Western blot.

**Sucrose gradient ultracentrifugation**. Cell extracts from mESCs were sedimented on a 10–30% sucrose gradient by centrifugation using an OPTIMA L-100XP rotor (Beckmann) at 247,605×g at 4 °C for 16 h. The gradient was fractioned and analyzed by SDS-PAGE. Indicated antibodies were used for Western blot.

**ChIP and bioinformatic analysis**. ChIP experiments were performed according to the procedure described previously[27], with modifications. For Flag ChIP, briefly, $3 \times 10^7$ cells were crosslinked with 2 mM EGS [ethylene glycol bis (succinimidylsuccinate)] for 45 min at RT with rotation at a speed of 8–10 rpm. Cells were washed with PBS twice and crosslinked again with 1% formaldehyde for 10 min at RT. Then the reaction was stopped by 0.125 M glycine. The cells were sonicated to achieve a chromatin sized of 200–500 bp in SDS lysis buffer containing 1 mM PMSF and protease inhibitor cocktail. Then the sonicated chromatin was dialyzed with TE buffer containing 150 mM NaCl and incubated with M2-Magnetic beads overnight at 4 °C with rotation. The complexes were washed twice with TBS-T (Tween-20 at 0.05%), and four times with TBS at 4 °C. For H2AK119ub1 and H3K27me3 ChIP, cells were washed with buffer A (10 mM Tris-HCl (pH7.9), 1.5 mM MgCl₂, 10 mM KCl, 0.1% Triton X-100, protease inhibitors). The resultant chromatin was re-suspended in 200 µl Buffer B (50 mM Tris–HCl (pH7.9), 3.5 mM MgCl₂, 2 mM CaCl₂, protease inhibitors) and followed by adding 800 µl of Buffer C (10 mM Tris–HCl (pH7.9), 25% glycerol, 1.5 mM MgCl₂, 440 mM NaCl, protease inhibitors). After rotation at 4 °C for 10 min, the chromatin was isolated by centrifuging at 1000×g at 4 °C for 5 min, washed twice with 200 µl buffer B and 800 µl Buffer C mixture again. The resultant chromatin was re-suspended in 200 µl Buffer B and digested into nucleosome fragments at 37 °C by adding 5 µl 200 U/ml MNase. Reaction was stopped by adding EGTA to 15 mM. After that, nucleosome fragments were crosslinked with 0.5% formaldehyde immediately. Incubate samples on a rotating wheel for 10 min at RT and stop the crosslink by adding glycine. ChIP reactions consisted of 50 µg of chromatin, 5 µg antibody, dynabeads protein A and G (1:1 mixed) and 5 µg of *Drosophila* chromatin spike-in for correct quantification to a final volume of 700 µl. ChIP reactions were incubated at 4 °C overnight with rotation. After incubation, immune complexes were washed with the following buffers: low salt wash buffer (0.1% SDS, 1% Triton X-100, 2 mM EDTA, 20 mM Tris-HCl (pH 8.0), 150 mM NaCl), high salt wash buffer (0.1% SDS, 1% Triton X-100, 2 mM EDTA, 20 mM Tris-HCl (pH 8.0), 500 mM NaCl), LiCl wash buffer (0.25 M LiCl, 1% IGEPAL-CA630, 1% deoxycholic acid (sodium salt),

1 mM EDTA, 10 mM Tris-HCl (pH 8.0)) and TE buffer (10 mM Tris-HCl (pH 8.0), 1 mM EDTA). Antibody-bound chromatin was reverse-crosslinked, and the ChIPed DNA samples were purified for either ChIP-Seq or qPCR amplification. The amounts of immunoprecipitated DNA were normalized to the input. The primers used in the qPCR assays are listed in Supplementary Table 4. All the experiments were repeated for three times.

About 10 ng IPed DNA and input DNA measured by Qubit Fluorometer (Invitrogen) were used to construct DNA library by using ChIP-seq Sample Prep Kit (Illumina). DNA libraries were sequenced and the raw reads were processed with the protocols adopted in previous study[28]. Briefly, the adapter and low quality sequences were trimmed from 3′ to 5′ ends by Trimmomatic[29]. After trimming, reads shorter than 36 bp were discarded. Subsequently, the preprocessed reads were aligned to mouse reference genome (mm9) using Bowtie2[30]. Then, the aligned reads were converted to bam format using samtools[31] and duplicates were removed by Picard (http://broadinstitute.github.io/picard). Finally, the histone-enriched regions were called by MACS2[32] with default parameters. For PCGF5 ChIP-seq, enriched peaks were called by MACS2 with p-value $<10^{-4}$ as cutoff, then peaks with q-value less than 0.01 were chosen for further analysis. Peaks were assigned to the nearby genes by annotatePeaks.pl function in the homer package[33]. Candidate target genes were identified if the peaks were located within ±2.5 kb of their TSSs. The density of histone signal was normalized with internal control as described[34], then visualized by UCSC genome browser[35]. All tag density heatmaps plots were created by homer and in the R package[36].

**RNA-seq and bioinformatic analysis**. To get high quality RNA-seq reads, raw RNA-seq data were subjected to a processing procedure: (1) Trimming adapter sequence from 3′ to 5′ end; (2) Trimming nucleotide with Phred quality score <5 from 3′ to 5′; (3) read with length less than 19 bps were removed; (4) If two reads were exactly identical to each other, the duplicated read was removed; (5) Remove reads with number of N bases accounting for more than 5%. Then, the remaining paired-end reads were mapped to mouse genome (mm9 assembly) using TopHat2[37], and gene expression levels were determined by cufflinks in the form of fragments per kilobase of exon per million fragments mapped (FPKM). For differentially expressed gene analysis, we calculated log2 fold change value (log2FC) for each gene in paired $Pcgf5^{-/-}$ and wild type samples. Upregulated and downregulated genes were selected using 1.5-fold change cutoff, and only genes with a mean RPKM value >1 in at least one condition were included. For functional enrichment analysis, all genes were then ranked by log2FC and used in a weighted, pre-ranked GSEA analysis[38] against a collection of gene sets from MSigDB and user defined gene sets by using the neurectoderm and mESC-specific gene lists[39]. Significant associations were determined for any gene set having a FWER p-value below 0.001.

**Quantification and statistical analysis**. Data are presented as mean values ± SD unless otherwise indicated in figure legends. Sample number (n) indicates the number of independent biological samples in each experiment. Sample numbers and experimental repeats are indicated in figures and figure legends or methods section above. Data were analyzed using Student's t-test analysis. Differences in means were considered statistically significant at $p < 0.05$. Significance levels are: *$p < 0.05$; **$p < 0.01$; ***$p < 0.001$.

**Data availability**. All sequencing data generated in this paper have been deposited in the NCBI. The GEO number for RNA-seq data is: GSE95127; GEO number for ChIP-seq data is: GSE107377. We have deposited the raw data for the qRT-PCR and ChIP-qPCR experiments into figshare (https://doi.org/10.6084/m9.figshare.5909101). The authors declare that all the data supporting the findings of this study are available within the article and its supplementary information files, or from the corresponding author upon reasonable request.

**Fig. 5** PCGF5 knockout impairs repressive chromatin from being reduced during neural differentiation. **a** Tag density heatmaps illustrating global changes of H2AK119ub1 in wild type and PCGF5-deficent ESCs. **b** Tag density heatmaps illustrating global changes of H3K27me3 in wild type and PCGF5-deficent ESCs. **c** Histone H2AK119ub1 occupancy (ChIP-seq read density) at three clusters of genes in both wild type and PCGF5-deficient mESCs and NPCs. **d** Histone H3K27me3 occupancy (ChIP-seq read density) at three clusters of genes in both wild type and PCGF5-deficient mESCs and NPCs. **e** GO analysis of biological functions of mESCs H2AK119ub1-enriched peaks. Results are expressed as −log10 (p-value). **f** GO analysis of biological functions of mESCs H3K27me3-enriched peaks. Results are expressed as −log10 (p-value). **g** UCSC genome browser views of binding profiles of H2AK119ub1 and H3K27me3 at the genes of Sox1, Nestin, Cdh2 and Pou3f2 in both wild type and PCGF5-deficient NPCs. **h** ChIP-qPCR analysis of PCGF5 occupancies at promoter regions of Sox1, Nestin, Cdh2 and Pou3f2 in both wild type and PCGF5-deficient mESCs and NPCs (at day 6 of neural differentiation) (n = 3). **i** ChIP-qPCR analysis of H2AK119ub1 occupancies at promoter regions of Sox1, Nestin, Cdh2 and Pou3f2 in both wild type and PCGF5-deficient mESCs and NPCs (at day 6 of neural differentiation) (n = 3). **j** ChIP-qPCR analysis of H3K27me3 occupancies at promoter regions of Sox1, Nestin, Cdh2 and Pou3f2 in both wild type and PCGF5-deficient mESCs and NPCs (at day 6 of neural differentiation) (n = 3). Data in **h–j** are represented as mean values ± s.d. with the indicated significance from Student's t-test (*p < 0.05, **p < 0.01, ***p < 0.001)

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

## Acknowledgements

We thank Dr. Xudong Wu for helpful discussion. We thank Dr. Guohong Li for providing histone octamers, Dr. Jicheng Zhao for helping with histone H2AK119ub1 ChIP experiments, Dr. Bing Zhu for providing Drosophila S2 Cells, Dr. Rui-Ming Xu for providing GST-PCGF4 construct, Dr. Naihe Jing for providing Sox1-GFP mESC lines, Dr. Jing Liu for advice. We also thank Mr. Taifeng Jiang for generating TALEN constructs. This work was supported by the Strategic Priority Research Program of the Chinese Academy of Sciences (XDA16010206), the Ministry of Science and Technology of the People's Republic of China (2016YFA0100400, 2015CB964800, 2016YFA0100300), National Science and Technology Major Projects for "New Drugs Innovation and Development" (2018ZX09201002-005), National Natural Science Foundation of China (31471210, 31601050, 31701131), Guangdong Frontier and Key Technology Innovation Special Grant (2016B030229006), Guangdong Natural Science Funds (2015A030308003, 2015A030310041, 2016A030313168), Guangzhou Science Technology and Innovation Commission (201807010101, 201707020042), General Research Funds (GRF) from the Research Grants Council (RGC) of the Hong Kong Special Administrative Region (14102315, 14113514, 14100415, 14116014), Focused Innovations Scheme (1907307). The authors also gratefully thank the support from the Guangzhou Branch of the Supercomputing Center of CAS.

## Authors contributions

H.Y and M.Y. initiated the study and designed the experiments. M.Y. performed most of the experiments. M.Y., X. Z., S.G., J.Li., K.H., P.L. and G.S. conducted the experiments. J. Z. and G.H. performed bioinformatics analysis. All other authors contributed to the work. H.Y. and M.Y. wrote the manuscript. H.Y. conceived and supervised the entire study.

## Additional information

**Competing interests:** The authors declare no competing interests.

