## [Peer Review File · Nature Communications]

Reviewers' comments:

Reviewer #1 (Remarks to the Author):

In this manuscript entitled "PCGF5 regulates neural differentiation of embryonic stem cells by modulating Nodal/TGF- β signaling pathway", Yao et al. describe the biochemical and cellular functions of polycomb group ring finger 5 (PCGF5), one subunit of Polycomb repressor complex 1 (PRC1) complexes in mESC neural differentiation. This study provides an innovative and important view of PCGF5 and the involved Nodal/TGF-beta signaling pathway in neural differentiation of mESCs. In general, the data and proposed mechanism presented in this study are highly interesting and novel, however, in my opinion, the authors should do some extra works in order to improve this manuscript and confirm their proposed model and mechanistic linkage. In particular, the points listed below should be addressed before this work can be considered for publication.

A few specific comments are attached as below:

1. The authors have showed that Pcgf5 promotes monoubiquitination of H2AK119 by Ring1B, in addition, they have showed that Pcgf5 loss of function results in reduction of ubH2A enrichment on the promoter of Nodal gene. However, the authors should perform ChIP-seq experiments in the not only mESCs but also PCGF5^{-/-} mESCs, even in the neural differentiation process at the early stage to see Pcgf5 loss of function on the global effects and distribution of histone H2AK119ub1 level in both mESCs and differentiated cells.
2. The discussion part is too short. Nodal/TGF- β signaling pathway has been previously reported and has a negative effect in regulating neural differentiation, while this manuscript mainly focus on the upstream of pcgf5 loss-of-fuction interfere with ESC neural differentiation by activating SMAD2 phosphorylation and the Nodal/TGF-beta signaling pathway. As the members of Pcgfs, except Pcgf5, the other members of pcgfs, such as pcgf1, pcgf2, pcgf6 have been previously implicated in ESC differentiation (Ross et al., Blood 2012; Morey et al., Cell stem Cell 2015; Zhao et al., The Journal of Biological Chemistry 2017), but these papers were neither cited nor discussed.
3. The manuscript has a very short background as well, more like a Nature Letter, it should be enlarged here.
4. RNA-seq has been performed in this manuscript. However, there is no GEO number contained. GEO number should be added in a revised version of manuscript for RNA-seq experiments.
5. In supplementary figure 1a, capital letters for different epigenetic genes should be used as small letters.
6. It is hard to assess how many technical/experimental/biological replicates experiments have been done for some experiments. The authors could add this information into the figure legends.
7. The authors should explain how they generated Pcgf5-depleted 46C mESCs and they should provide the detailed information in the paper.

Reviewer #2 (Remarks to the Author):

In this manuscript, Yao and co-authors dissect the role of the non-canonical Polycomb Repressive Complex 1 member Pcgf5 in the regulation of neural differentiation of embryonic stem (ES) cells. They used the TALEN gene editing system to generate Pcgf5 knockout ES cells. The loss of Pcgf5 did not lead to any change in ES cell maintenance or the expression of any of the key pluripotency factors. However, intriguingly the Pcgf5 null cells displayed a failure to silence these pluripotency factors and to activate neural specific genes during induced neural differentiation. They go on to link this block in neural differentiation to an increase in TGFB pathway activity. They show that inhibition of the TGFB pathway in Pcgf5 null cells partially rescues the neural differentiation phenotype; however the rescue is not very strong. Finally they find that there is a reduction in Polycomb activity at the Nodal promoter in Pcgf5 null cells and

this correlates with the observed increase in Nodal transcription during neural differentiation.

Major points:

1. The partial rescue of Sox1, Nestin and Pax6 gene expression upon TGFB pathway inhibition in Figure 2C is mild at best. This would suggest that the role of hyper active TGFB pathway in Pcgf5 null cells is not actually causing the neural differentiation defect.
2. In Figure 2F and 2G, inhibition of the TGFB pathway by LY2109761 in wild type cells seems to lead to an increase in expression of neural markers (Nestin, Pax6 and Sox1). This suggests that the observed increase in expression of these same genes upon treatment with LY2109761 is solely a result of TGFB pathway inhibition and is independent of the loss of Pcgf5.
3. The authors should perform Western blots of H2AK119ub1 and H3K27me3 in Pcgf5 null ES and NSC. If Pcgf5 is essential for the presence of these histone marks in either cell type then you may expect a change in global levels of these marks. This would reinforce the ChIP evidence for a reduction in these marks at the Nodal promoter in figure 4.
4. In addition to Western blotting, the authors should also carry out ChIP-RX (with a drosophila chromatin spike-in for correct quantification) of H2AK119ub1 and H3K27me3 in ES and NSC. This would give a more comprehensive overview of the role of Pcgf5 in mediating these histone modifications. The analysis presented in the manuscript only focusses on one gene (Nodal), which is not a thorough approach to determine the effect of Pcgf5 loss on histone modifications at the other thousands of Polycomb target genes.
5. The ChIP evidence presented in figure 4 suggests that Pcgf5 has a role in repressing Polycomb targets (albeit only at one gene). A previous report (Gao et al, Nature, 2014) provided evidence that Pcgf5 functions as an activator of transcription following artificial recruitment to the luciferase promoter. For this reason the authors should perform Pcgf5 ChIP seq to determine its direct target genes. This will allow a proper assessment of the direct role of Pcgf5 in transcription and mediation of H2AK119ub1 and H3K27me3. The effects exhibited in figure 4 at the Nodal promoter may be an indirect loss of repression of Nodal caused by transcriptional changes at a different gene which could be the real Pcgf5 target. Therefore, coupling the existing RNA-seq analysis from figure 1 to a Pcgf5 ChIP-seq would go a long way to uncovering the genome-wide role of Pcgf5.
6. The model presented in figure 4J should be amended to reduce the focus on competition between Pcgf5 and the TGFB pathway, since this link does not seem to be all that strong. Instead the focus should be shifted to the role of Pcgf5 in regulating H2AK119ub1 and PRC2 recruitment/H3K27me3 deposition at Polycomb target genes and also illustrate the block in differentiation exhibited in the absence on Pcgf5. The authors have some very compelling preliminary evidence in Figure 1 particularly and I have made suggestions above on how to extend this.

Minor points

1. Figure 1B and 1C do not contribute to our understanding of Pcgf5 function since they only illustrate how the authors made the useful null cell line to study its function. Therefore these panels could be moved to a supplementary figure.
2. Figure 2F could be better presented as a bar chart showing the levels of Sox1 alone. The y-axis in this FACs analysis does not inform the reader.
3. The immunoprecipitation experiment presented in Figure 3A is not informative. Performing a Flag IP of Pcgf5 and then probing for H2A when the control IP does not have overexpressed H2A is not a properly controlled experiment.
4. The findings of figure 3B and figure 3J are already well established by many authors (such as Gao et al, Nature, 2014/ Gao et al, Molecular Cell, 2012/ Hauri et al, Cell Reports 2016) and therefore should not be included in the main figures.
5. The immunofluorescence images in figure 3J should be quantified by quantifying Pcgf5 and H2AK119ub1 in cells in several fields and then quantifying the correlation between high Pcgf5 and high H2AK119ub1.
6. In Figures 3C-F, the authors nicely map the interaction regions between Ring1B and Pcgf5. To boost the ubiquitination assay presented in Figure 3I, they could include a mutant Pcgf5 incapable of interaction with Ring1B. One would expect that the Ring finger mutant of Pcgf5

would have greatly reduced ubiquitination activity.

7. In the figure 4 title, the authors use the phrase "Loss of Pcgf5 reduces the recruitment of histone H2AK119ub1 onto the promoter of nodal...". Histone modifications are not recruited, they are catalysed in situ, and therefore this title should be rephrased to reflect this.

8. Some nice data is left in the supplementary figure and could be moved to the main figure. The RT-PCRs in FS1A show that Pcgf5 is highly up-regulated during neural differentiation. It is the only non-canonical Pcgf that seems highly expressed in NSCs. The quantification of Pcgf1-6 expression in this supplementary figure could be moved to the main figure 1 to illustrate the potential importance of Pcgf5 in NSC prior to the authors characterising their Pcgf5 null cells.

Response to Reviewers' comments:

Reviewers' comments:

Reviewer #1 (Remarks to the Author):

In this manuscript entitled "PCGF5 regulates neural differentiation of embryonic stem cells by modulating Nodal/TGF- β signaling pathway", Yao et al. describe the biochemical and cellular functions of polycomb group ring finger 5 (PCGF5), one subunit of Polycomb repressor complex 1 (PRC1) complexes in mESC neural differentiation. This study provides an innovative and important view of PCGF5 and the involved Nodal/TGF-beta signaling pathway in neural differentiation of mESCs. In general, the data and proposed mechanism presented in this study are highly interesting and novel, however, in my opinion, the authors should do some extra works in order to improve this manuscript and confirm their proposed model and mechanistic linkage. In particular, the points listed below should be addressed before this work can be considered for publication.

A few specific comments are attached as below:

1. The authors have showed that Pcgf5 promotes monoubiquitination of H2AK119 by Ring1B, in addition, they have showed that Pcgf5 loss of function results in reduction of ubH2A enrichment on the promoter of Nodal gene. However, the authors should perform ChIP-seq experiments in the not only mESCs but also PCGF5^{-/-} mESCs, even in the neural differentiation process at the early stage to see Pcgf5 loss of function on the global effects and distribution of histone H2AK119ub1 level in both mESCs and differentiated cells.

Answer: We followed reviewer's suggestion and performed ChIP-seq experiments by using anti-H2AK119ub1, anti-H3K27me3 antibodies in both wild type and PCGF5-depleted mESCs and NPCs (at day 6 of mESC neural differentiation). Our results indicated that PCGF5 loss-of-function decreases the level of histone

H2AK119ub1 and H3K27me3 at the TSS regions of Nodal, Lefty1 and Lefty that are TGF-β associated genes compared with wild type NPCs, which is consistent with H2AK119ub1 and H3K27me3 ChIP-qPCR results. We added these ChIP-seq data into Figure 4 and Figure 5, as well as supplementary figures 7-9.

2. The discussion part is too short. Nodal/TGF-β signaling pathway has been previously reported and has a negative effect in regulating neural differentiation, while this manuscript mainly focus on the upstream of pcgf5 loss-of-function interfere with ESC neural differentiation by activating SMAD2 phosphorylation and the Nodal/TGF-beta signaling pathway. As the members of Pcgfs, except Pcgf5, the other members of pcgfs, such as pcgf1, pcgf2, pcgf6 have been previously implicated in ESC differentiation (Ross et al., Blood 2012; Morey et al., Cell stem Cell 2015; Zhao et al., The Journal of Biological Chemistry 2017), but these papers were neither cited nor discussed.

Answer: We have followed the reviewer's suggestions and cited these papers and added the following sentences into the discussion part.

"Pcgf1/NSPc1, together with the transcription factor Runx1, regulate HSC differentiation and self-renewal. PCGF2, also known as Mel-18, positively regulates expression of key mesoderm transcription factors, revealing an unexpected function in gene activation during cardiac differentiation."

"Pcgf6 deletion causes a dramatic decrease in PRC1.6 binding to target genes and no loss of H2AK119ub1¹. Similarly, recent observations have shown that the global level of H2AK119ub1 is unchanged in Pcgf1^{-/-} ES cells². These alterations in molecular activity of PRC1 lead to changes in developmental outcome."

"PCGF5 has been demonstrated to contribute to H2AK119ub1-dependent recruitment of PRC2 and H3K27me3 modification in a manner similar to other non-canonical PRC1 complexes in mESCs³."

We also added the following sentences into the discussion.

" Previous reports revealed that PCGF5-PRC1-AUTS2 complex is involved in gene activation by transcriptional co-activator p300 after Ring1B is phosphorylated by the CK2 protein4. We investigated if PCGF5 co-binds to specific targets with histone repressive marks, H2AK119ub1, H3K27me3 and active marks H3K27ac5 at day 6 after mESC neural differentiation. Our data indicated that only a small fraction of PCGF5 target genes overlapped with the repressive H2AK119ub1- and H3K27me3-enriched genes (Supplementary Fig. 9a). However, we surprisingly found that most PCGF5 target genes overlapped with the genome-wide localizations of active marks H3K27ac- and H3K4me35 in NPCs (Supplementary Fig. 9b). Our data further indicated that about 39.5% (1024) of PCGF5-binding sites target to the active genes and only 6.6% (170) of PCGF5-binding sites target silent genes (Supplementary Fig. 9c). These data suggest that PCGF5 may function as an activator of NPC-specific genes during mESC neural differentiation.

Our data suggest that PCGF5 not only acts to maintain the levels of H2AK119ub1 and H3K27me3 at key TGF- β signaling genes, and that this is required for the repression of these genes, but also facilitates the reduction of H2AK119ub1 and H3K27me3 around the promoters of neural-specific genes during neural differentiation, suggesting PCGF5 plays dual functions in regulating mESC neural differentiation, acting as a repressor for TGF- β signaling pathway and functioning as an activator for neural-related genes in another unknown mechanism."

3. The manuscript has a very short background as well, more like a Nature Letter, it should be enlarged here.

Answer: We have enlarged the introduction by following reviewer's suggestions.

4. RNA-seq has been performed in this manuscript. However, there is no GEO number contained. GEO number should be added in a revised version of manuscript for RNA-seq experiments.

Answer: We added GEO number for both RNA-seq data and ChIP-seq data into the

newly updated manuscript. The GEO number for RNA-seq data is: GSE95127; GEO number for ChIP-seq data is: GSE107377.

5. In supplementary figure 1a, capital letters for different epigenetic genes should be used as small letters.

Answer: We sincerely accept reviewer's comments and have revised the capital letters as small letters in figure 1a (previous as supplementary figure 1a).

6. It is hard to assess how many technical/experimental/biological replicates experiments have been done for some experiments. The authors could add this information into the figure legends.

Answer: Actually, in this manuscript, all experiments that we showed in this manuscript were repeated for three times. We appreciated the reviewer's suggestions and provided the n=3 in the figure legends.

7. The authors should explain how they generated Pcgf5-depleted 46C mESCs and they should provide the detailed information in the paper.

Answer: We added the detailed procedure on how we generated Pcgf5-depleted 46C mESCs into this manuscript as below.

"We used the RNAi lentivirus system for generation of PCGF5-depleted stable 46C mESC lines. In brief, shRNA sequence targeting mouse PCGF5 was cloned into pLKO.1, according to protocols described online

(<http://www.addgene.org/tools/protocols/plko/#E>). The recombinant construct, as well as two assistant vectors psPAX2 (Addgene #12260) and MD2.G (Addgene #1225), were co-transfected into HEK293T cells. Viral supernatants were collected 48 h later, filtrated through 0.45 μ m filters. The viruses were used to infect 5×10^6 mESCs in a 6 cm dish with 8 μ g/mL polybrene. The clonal cell clusters of mESCs were picked out

with micro-needle after 2 $\mu\text{g/mL}$ puromycin (Amresco) selection."

Reviewer #2 (Remarks to the Author):

In this manuscript, Yao and co-authors dissect the role of the non-canonical Polycomb Repressive Complex 1 member Pcgf5 in the regulation of neural differentiation of embryonic stem (ES) cells. They used the TALEN gene editing system to generate Pcgf5 knockout ES cells. The loss of Pcgf5 did not lead to any change in ES cell maintenance or the expression of any of the key pluripotency factors. However, intriguingly the Pcgf5 null cells displayed a failure to silence these pluripotency factors and to activate neural specific genes during induced neural differentiation. They go on to link this block in neural differentiation to an increase in TGFB pathway activity. They show that inhibition of the TGFB pathway in Pcgf5 null cells partially rescues the neural differentiation phenotype; however the rescue is not very strong. Finally they find that there is a reduction in Polycomb activity at the Nodal promoter in Pcgf5 null cells and this correlates with the observed increase in Nodal transcription during neural differentiation.

Answer: We thank the reviewer#2's comments on this manuscript. We have performed more experiments and address the questions as below.

Major points:

1. The partial rescue of Sox1, Nestin and Pax6 gene expression upon TGFB pathway inhibition in Figure 2C is mild at best. This would suggest that the role of hyper active TGFB pathway in Pcgf5 null cells is not actually causing the neural differentiation defect.

Answer: We repeated this experiment again and agreed with the reviewer's comments that the partial rescue of neural differentiation defect caused by loss of PCGF5 is mild. However, we indeed discovered that PCGF5 loss-of-function resulted in activation of SMAD2/TGF- β signaling pathway by the increase of SMAD2 phosphorylation (Fig. 2d). Furthermore, considering question 4 that was raised by the reviewer, we

performed ChIP-seq for H2AK119ub1, H3K27me3 in both wild type and PCGF5-depleted mESCs and NPCs, we found that PCGF5 loss-of-function not only resulted in lower level of H2AK119ub1, H3K27me3 at the promoters of TGF- β signaling pathway genes (Nodal, Lefty1 and Lefty2), but also blocked the reduction of H2AK119ub1, H3K27me3 at the promoters of neural specific genes and kept them repressed. Therefore, PCGF5 might function as both a repressor for TGF- β signaling pathway, and a regulator for neural specific genes during mESC neural differentiation. We added these ChIP-seq data into this new manuscript. In addition, we removed SMAD2 phosphorylation data from Fig. 5 and added this data into Fig. 2 to make the manuscript more clear.

2. In Figure 2F and 2G, inhibition of the TGF β pathway by LY2109761 in wild type cells seems to lead to an increase in expression of neural markers (Nestin, Pax6 and Sox1). This suggests that the observed increase in expression of these same genes upon treatment with LY2109761 is solely a result of TGF β pathway inhibition and is independent of the loss of Pcgf5.

Answer: We appreciated the reviewer's comments. We further repeated this experiment more than three times, we found that inhibition of the TGF- β signaling pathway by LY2109761 in wild type cells indeed leads to an increase in expression of neural markers Pax6 and Nestin. However, we also found that inhibition of the TGF- β signaling pathway by LY2109761 in wild type cells has no effect on the expression of neural markers Sox1, an early marker for neural induction⁶, at both day 4 (see below Fig. 1a, b) and day 6 after mESC neural differentiation. In addition, previous study suggested that TGF- β signaling pathway inhibitor could not promote neural differentiation of mESCs⁷(See Fig. 1c from the cited paper 7 below). Therefore, we conclude that inhibition of TGF- β signaling pathway partially rescues Sox1 expression during neural differentiation upon PCGF5-loss-of-function.

For *Nestin* and *Pax6*, when we treated the cells with another TGF- β signaling pathway inhibitor SB431542, we could not see the increase of RNAs for both *Nestin* and *Pax6* in wild type cells during neural differentiation (see below Fig. 1d, e). Together, we speculate that selectivity of LY2109761 or other unknown mechanisms might result in the increase of gene expression for these two markers. We removed previous Fig.2f, g.

Figure 1. **a.** Gene expression analysis of Sox1 at day 4 after neural differentiation. DMSO, Ly2109761 was added into the media during neural differentiation (n=3). Results are shown relative to wild type at day 4. **b.** Statistical analysis of positive cells expressing Sox1-GFP described a. **c.** In mESCs, SB431542 had no positive effects on the production of neural stem cells (NSCs) at standard density. However, SB431542 prevents an inhibitory effect on neural induction mediated by high-density plating shown by ICC analysis (left panel) and quantified data (right panel). (Matulka et al., cell stem cell, 2013). **d, e.** Gene expression analysis of Pax6, Nestin at day 6 after neural differentiation. DMSO, SB431542 was added into the media during neural differentiation (n=3). Results are shown relative to wild type at day 6. The data in a, b, d, e are plotted from three independent experiments, each with three technical replicates. Data in a, b, d, e, are represented as mean values \pm s.d, with the indicated significance from Student's t test (**P<0.01, ***P<0.001, NS, no significant).

3. The authors should perform Western blots of H2AK119ub1 and H3K27me3 in Pcgf5 null ES and NSC. If Pcgf5 is essential for the presence of these histone marks in either cell type then you may expect a change in global levels of these marks. This would reinforce the ChIP evidence for a reduction in these marks at the Nodal promoter in figure 4.

Answer: By following the reviewer's suggestion, we performed Western blot for H2AK119ub1 and H3K27me3 in both wild-type and Pcgf5-depleted ESCs and NPCs. The result showed that PCGF5 loss-of-function slightly reduces H2AK119ub1 level but has no effects on H3K27me3 level in mESCs, although H3K27me3 level decreased during mESC neural differentiation in both wild-type and Pcgf5-depleted cells (Supplementary Fig.7a, b). These are consistent with previous reports that the abundance of H3K27me3 mark decreased during neural differentiation⁸. We would like to emphasize that it is a common observation that the loss of epigenetic regulators does not necessarily lead to an obvious change in the global level of an epigenetic

mark⁸, instead there are often relatively small context-specific changes, which we characterized with our ChIP-seq data.

4. In addition to Western blotting, the authors should also carry out ChIP-RX (with a drosophila chromatin spike-in for correct quantification) of H2AK119ub1 and H3K27me3 in ES and NSC. This would give a more comprehensive overview of the role of Pcgf5 in mediating these histone modifications. The analysis presented in the manuscript only focusses on one gene (Nodal), which is not a thorough approach to determine the effect of Pcgf5 loss on histone modifications at the other thousands of Polycomb target genes.

Answer: We followed the reviewer's great suggestions and performed ChIP-RX by using anti-H2AK119ub1 and anti-H3K27me3 antibodies in both wild-type and Pcgf5-depleted mESCs and NPCs (at day 6 of mESC differentiation). For correct quantification, we added 10% drosophila chromatin spike-in into the chromatin from either mESCs or NPCs. Our results showed that PCGF5 loss-of-function did not lead to global changes in both H2AK119ub1 and H3K27me3 in ESCs (see below Fig. 2a,b), therefore, we did not show these data in the manuscript; However, PCGF5 loss-of-function significantly inhibited the reduction of ESCs enriched peaks for both H2AK119ub1 and H3K27me3 in NPCs, respectively, including many neural-related genes, such as Sox1, Nestin, Cdh2. In addition, loss of PCGF5 resulted in lower levels of NPCs enriched peaks for both H2AK119ub1 and H3K27me3 in NPCs, including many genes that are highly expressed in ESCs relative to NPCs (such as Nodal).

Figure 2. Metaplots illustrating global changes in both H2AK119ub1 (a) and H3K27me3 (b) in Wild type and Pcgf5-depleted ESCs.

5. The ChIP evidence presented in figure 4 suggests that Pcgf5 has a role in repressing Polycomb targets (albeit only at one gene). A previous report (Gao et al, Nature, 2014) provided evidence that Pcgf5 functions as an activator of transcription following artificial recruitment to the luciferase promoter. For this reason the authors should perform Pcgf5 ChIP seq to determine its direct target genes. This will allow a proper assessment of the direct role of Pcgf5 in transcription and mediation of H2AK119ub1 and H3K27me3. The effects exhibited in figure 4 at the Nodal promoter may be an indirect loss of repression of Nodal caused by transcriptional changes at a different gene which could be the real Pcgf5 target. Therefore, coupling the existing RNA-seq analysis from figure 1 to a Pcgf5 ChIP-seq would go a long way to uncovering the genome-wide role of Pcgf5.

Answer: We have followed the reviewer's suggestions and performed PCGF5 ChIP-seq in NPCs at day 6 after neural differentiation. At the beginning, we performed ChIP-seq by using commercial anti-PCGF5 antibodies; unfortunately, these antibodies did not work for ChIP. Therefore, we generated Flag-tagged PCGF5 knockin stably expressing in Sox1-GFP mESCs (Fig. 4a-c and Supplementary Fig. 6a) and performed ChIP-seq by using anti-Flag antibodies in NPCs (at day 6 after neural differentiation). Our ChIP-seq data identified 12,015 peaks in NPCs for PCGF5 ChIP-seq analysis revealed a preferential distribution of PCGF5 near transcription start sites (TSS) of

genes in NPCs (Fig.4d). About 25.04% of PCGF5 sites are near promoter regions. 35.87% of the PCGF5-binding sites are located in the intergenic regions, a significant number of PCGF5-binding sites fall within genes, with 33.55% in the introns and 5.54 in the exons (Fig.4e). We further focused on the effects of PCGF5 on the genes in NPCs. Among 80 genes in the TGF- β signaling pathway, 35 of them were directly targeted by PCGF5.

According to the reviewer's nice comments that assessment of the direct role of Pcgf5 in transcription and mediation of H2AK119ub1 and H3K27me3. We performed ChIP-seq by using anti-H2AK119ub1 and anti-H3K27me3 antibodies in both wild type and Pcgf5-depleted NPCs. We investigated whether PCGF5 co-bound to specific genes with histone repressive marks, H2AK119ub1 and H3K27me3, we found that only small fraction of PCGF5-binding sites overlapped with repressive marks H2AK119ub1- and H3K27me3-binding sites (Supplementary Fig. 9a). However, our data indicated that most of PCGF5-binding sites overlapped with the published results genome-wide localizations of active marks H3K27ac- and H3K4me3-binding sites (Supplementary Fig. 9b). Interestingly, we further found that about 39.5% (1024) of PCGF5-binding sites targeted to the active genes and only 6.6% (170) of PCGF5-binding sites bound to silent genes (Supplementary Fig. 9c). PCGF5 loss-of-function resulted in 521 downregulated PCGF5 target genes that were associated with axon development, neuron project development (Fig. 4f,g). And the upregulated genes targeted by PCGF5 were involved in mRNA processing, ribonucleoprotein complex biogenesis (Fig. 4g). Included within these 99 upregulated PCGF5 target genes were Nodal, Lefty1 and Lefty2, three TGF- β signaling pathway genes. Hence, we performed ChIP-qPCR experiments to examine the recruitments of PCGF5, H2AK119ub1 and H3K27me3 onto the promoters of these genes. Our data indicated that PCGF5 was indeed specifically recruited to the promoters of Nodal, Lefty1 and Lefty2 at day 6 during neural differentiation of mESCs (Fig. 4h). On the other hand, PCGF5 loss-of-function

prevented the reduction of H2AK119ub1 and H3K27me3 around neural specific genes and kept them repressed (Fig. 5e-h), further suggesting that PCGF5 might be required for activation of NPC-related genes. We added these data into the newly updated manuscript.

6. The model presented in figure 4J should be amended to reduce the focus on competition between Pcgf5 and the TGFB pathway, since this link does not seem to be all that strong. Instead the focus should be shifted to the role of Pcgf5 in regulating H2AK119ub1 and PRC2 recruitment/H3K27me3 deposition at Polycomb target genes and also illustrate the block in differentiation exhibited in the absence on Pcgf5. The authors have some very compelling preliminary evidence in Figure 1 particularly and I have made suggestions above on how to extend this.

Answer: We highly appreciated this great suggestion. We performed more experiments and our data suggested that PCGF5 not only acts to maintain the levels of H2AK119ub1 and H3K27me3 at key TGF- β signaling genes, and that this is required for the repression of these genes, but also facilitates the reduction of H2AK119ub1 and H3K27me3 around the promoters of neural-specific genes during neural differentiation, suggesting PCGF5 might play dual functions in regulating mESC neural differentiation, acting as a repressor for TGF- β signaling pathway and functioning as a facilitator for neural-related genes.

Minor points

1. Figure 1B and 1C do not contribute to our understanding of Pcgf5 function since they only illustrate how the authors made the useful null cell line to study its function.

Therefore these panels could be moved to a supplementary figure.

Answer: We have followed the reviewer's suggestion and moved Figure 1B and 1C to

Supplementary Fig. 1f, g.

2. Figure 2F could be better presented as a bar chart showing the levels of Sox1 alone. The y-axis in this FACs analysis does not inform the reader.

Answer: We followed the reviewer's suggestion and added Fig. 2h (previous as Figure 2f) with a bar chart showing Sox1 levels in different treatments in Fig. 2i.

3. The immunoprecipitation experiment presented in Figure 3A is not informative. Performing a Flag IP of Pcgf5 and then probing for H2A when the control IP does not have overexpressed H2A is not a properly controlled experiment.

Answer: We have redone this experiment and performed Flag co-IP experiment by using endogenous H2A (Fig. 3a).

4. The findings of figure 3B and figure 3J are already well established by many authors (such as Gao et al, Nature, 2014/ Gao et al, Molecular Cell, 2012/ Hauri et al, Cell Reports 2016) and therefore should not be included in the main figures.

Answer: We followed the reviewer's suggestions and moved Figure 3b and 3j to supplementary Fig. 5a, b.

5. The immunofluorescence images in figure 3J should be quantified by quantifying Pcgf5 and H2AK119ub1 in cells in several fields and then quantifying the correlation between high Pcgf5 and high H2AK119ub1.

Answer: We have followed the reviewer's suggestion and quantified PCGF5 and H2AK119ub1 levels in cells in multiple fields. We also analyzed the correlation between high Pcgf5 and high H2AK119ub1. We added these data into supplementary Fig. 5c, d.

6. In Figures 3C-F, the authors nicely map the interaction regions between Ring1B and Pcgf5. To boost the ubiquitination assay presented in Figure 3I, they could include a mutant Pcgf5 incapable of interaction with Ring1B. One would expect that the Ring finger mutant of Pcgf5 would have greatly reduced ubiquitination activity.

Answer: We sincerely accepted reviewer's comments. We further performed in vitro ubiquitination assay and found that histone H2A ubiquitination activity have greatly reduced with a mutant Pcgf5 without Ring-finger. We added this data into Fig. 3i.

7. In the figure 4 title, the authors use the phrase "Loss of Pcgf5 reduces the recruitment of histone H2AK119ub1 onto the promoter of nodal...". Histone modifications are not recruited, they are catalysed in situ, and therefore this title should be rephrased to reflect this.

Answer: We appreciate this suggestion. We have changed this sentence from "Loss of Pcgf5 reduces the recruitment of histone H2AK119ub1 onto the promoter of nodal...." to "loss-of-function reduces the deposition of histone H2AK119ub1 onto the promoters of TGF- β target genes."

8. Some nice data is left in the supplementary figure and could be moved to the main figure. The RT-PCRs in FS1A show that Pcgf5 is highly up-regulated during neural differentiation. It is the only non-canonical Pcgf that seems highly expressed in NSCs. The quantification of Pcgf1-6 expression in this supplementary figure could be moved to the main figure 1 to illustrate the potential importance of Pcgf5 in NSC prior to the authors characterising their Pcgf5 null cells.

Answer: We have followed reviewer's suggestion and we have moved supplementary Fig.1a to Fig.1a.

References

- 1 Zhao, W. *et al.* Essential Role for Polycomb Group Protein Pcgf6 in Embryonic Stem Cell Maintenance and a Noncanonical Polycomb Repressive Complex 1 (PRC1) Integrity. *The Journal of biological chemistry* **292**, 2773-2784, doi:10.1074/jbc.M116.763961 (2017).
- 2 Yan, Y. *et al.* Loss of Polycomb Group Protein Pcgf1 Severely Compromises Proper Differentiation of Embryonic Stem Cells. *Sci Rep* **7**, 46276, doi:10.1038/srep46276 (2017).
- 3 Blackledge, N. P. *et al.* Variant PRC1 complex-dependent H2A ubiquitylation drives PRC2 recruitment and polycomb domain formation. *Cell* **157**, 1445-1459, doi:10.1016/j.cell.2014.05.004 (2014).
- 4 Gao, Z. *et al.* An AUTS2-Polycomb complex activates gene expression in the CNS. *Nature* **516**, 349-354, doi:10.1038/nature13921 (2014).
- 5 Bonev, B. *et al.* Multiscale 3D Genome Rewiring during Mouse Neural Development. *Cell* **171**, 557-572 e524, doi:10.1016/j.cell.2017.09.043 (2017).
- 6 Wood, H. B. & Episkopou, V. Comparative expression of the mouse Sox1, Sox2 and Sox3 genes from pre-gastrulation to early somite stages. *Mech Dev* **86**, 197-201 (1999).
- 7 Matulka, K. *et al.* PTP1B is an effector of activin signaling and regulates neural specification of embryonic stem cells. *Cell stem cell* **13**, 706-719, doi:10.1016/j.stem.2013.09.016 (2013).
- 8 Kloet, S. L. *et al.* The dynamic interactome and genomic targets of Polycomb complexes during stem-cell differentiation. *Nature structural & molecular biology* **23**, 682-690, doi:10.1038/nsmb.3248 (2016).

REVIEWERS' COMMENTS:

Reviewer #1 (Remarks to the Author):

The authors have conducted additional experiments and successfully addressed all the concerns raised by this reviewer and the current form of this revised manuscript is acceptable.

Reviewer #2 (Remarks to the Author):

Yao et al, have taken on my original comments very well which has strengthened the revised manuscript. Here are some minor changes still needed.

One further change that could be made to simplify the message of the paper for the reader is to show heatmaps (similar to figure 5A and B) of the FLAG-PCGF5 ChIP-seq aligned alongside the H3K27me3 and H2AK119ub1 ChIP-seq results in ESCs and NPCs, both in WT and PCGF5 null cells. This would be very helpful to help the reader delineate the direct effects that loss of PCGF5 has at its own target genes from indirect effects happening at other genes. To this end, it would also help if the FLAG-PCGF5 ChIP seq tracks were also included in Figure 5E.

In addition, the nice metagene plots for H3K27me3 and H2AK119ub1 in WT and PCGF5 null cells provided in the rebuttal letter should be included in the paper to give the reader a good overall view of what effect loss of PCGF5 has.

The title of Figure 5 should be toned down from "PCGF5 loss of function prevents the reduction of H2AK119ub1 and H3K27me3 at the promoters of neural-related genes during neural differentiation" to "PCGF5 loss of function impairs the reduction of H2AK119ub1 and H3K27me3 at the promoters of neural-related genes during neural differentiation".

The initial manuscript had correct capitalization of letters for the current figure 1A. As these are human genes all letters should be capitalized.

The newly added sentence in the discussion 'Pcgf6 deletion causes a dramatic decrease in PRC1.6 binding to target genes and no loss of H2AK119ub1' is a little misleading, since both Zhao et al, 2017 (JBC) and Endoh et al, 2017 (Elife) report that in the absence of Pcgf6, its target genes had specific losses of H2AK119ub1 while other PRC1 target genes did not lose any H2AK119ub1. Therefore, this sentence should be rephrased to reflect this and also the Endoh et al, 2017, Elife reference should be added.

Response to Reviewers' comments

REVIEWERS' COMMENTS:

Reviewer #1 (Remarks to the Author):

The authors have conducted additional experiments and successfully addressed all the concerns raised by this reviewer and the current form of this revised manuscript is acceptable.

Response: We thank the reviewer for carefully examining our study.

Reviewer #2 (Remarks to the Author):

Yao et al, have taken on my original comments very well which has strengthened the revised manuscript. Here are some minor changes still needed.

Question 1. One further change that could be made to simplify the message of the paper for the reader is to show heatmaps (similar to figure 5A and B) of the FLAG-PCGF5 ChIP-seq aligned alongside the H3K27me3 and H2AK119ub1 ChIP-seq results in ESCs and NPCs, both in WT and PCGF5 null cells. This would be very helpful to help the reader delineate the direct effects that loss of PCGF5 has at its own target genes from indirect effects happening at other genes. To this end, it would also help if the FLAG-PCGF5 ChIP seq tracks were also included in Figure 5E.

Answer: *Following reviewer 2's suggestion, we have added heatmaps for Pcgf5 ChIP-seq data in both ESCs and NPCs into Fig. 5c and 5d*

(previous Figure 5a,b). We have also added PCGF5 ChIP-seq track for Sox1, Cdh2, Pou3f2 and Nestin into Figure 5g (previous Figure 5e).

Question 2. In addition, the nice metagene plots for H3K27me3 and H2AK119ub1 in WT and PCGF5 null cells provided in the rebuttal letter should be included in the paper to give the reader a good overall view of what effect loss of PCGF5 has.

Answer: We have added metagene density heatmap plots for H3K27me3 and H2AK119ub1 in WT and PCGF5 null cells into Figure 5a and 5b by following the second reviewer's suggestions.

Question 3. The title of Figure 5 should be toned down from "PCGF5 loss of function prevents the reduction of H2AK119ub1 and H3K27me3 at the promoters of neural-related genes during neural differentiation" to "PCGF5 loss of function impairs the reduction of H2AK119ub1 and H3K27me3 at the promoters of neural-related genes during neural differentiation".

Answer: We thank this reviewer's suggestion. We have changed the title of Figure 5 from "PCGF5 loss of function prevents the reduction of H2AK119ub1 and H3K27me3 at the promoters of neural-related genes during neural differentiation" to "PCGF5 knockout impairs repressive chromatin from being reduced at neural genes during neural differentiation."

Question 4. The initial manuscript had correct capitalization of letters for the current figure 1A. As these are human genes all letters should be capitalized.

Answer: We have capitalized the human gene names in Figure 1A.

Question 5. The newly added sentence in the discussion 'Pcgf6 deletion causes a dramatic decrease in PRC1.6 binding to target genes and no loss of H2AK119ub1' is a little misleading, since both Zhao et al, 2017 (JBC) and Endoh et al, 2017 (Elife) report that in the absence of Pcgf6, its target genes had specific losses of H2AK119ub1 while other PRC1 target genes did not lose any H2AK119ub1. Therefore, this sentence should be rephrased to reflect this and also the Endoh et al, 2017, Elife reference should be added.

Answer: We have rephrased the sentence in question to "Previous reports showed that in the absence of Pcgf6, its target genes had specific losses of H2AK119ub1 while other PRC1 target genes did not lose any H2AK119ub1". In addition, we have added the two references (Zhao et al., JBC, 2017; Endoh et al., Elife, 2017) as recommended.